# Provably Efficient Online RLHF with One-Pass Reward Modeling

**Long-Fei Li**[*], **Yu-Yang Qian**[*], **Peng Zhao, Zhi-Hua Zhou**
National Key Laboratory for Novel Software Technology, Nanjing University, China
School of Artificial Intelligence, Nanjing University, China
{lilf, qianyy, zhaop, zhouzh}@lamda.nju.edu.cn

## Abstract

Reinforcement Learning from Human Feedback (RLHF) has shown remarkable success in aligning Large Language Models (LLMs) with human preferences. Traditional RLHF methods rely on a fixed dataset, which often suffers from limited coverage. To this end, online RLHF has emerged as a promising direction, enabling iterative data collection and refinement. Despite its potential, this paradigm faces a key bottleneck: the requirement to continuously integrate new data into the dataset and re-optimize the model from scratch at each iteration, resulting in computational and storage costs that grow linearly with the number of iterations. In this work, we address this challenge by proposing a *one-pass* reward modeling method that eliminates the need to store historical data and achieves constant-time updates per iteration. Specifically, we first formalize RLHF as a contextual preference bandit and develop a new algorithm based on online mirror descent with a tailored local norm, replacing the standard maximum likelihood estimation for reward modeling. We then apply it to various online RLHF settings, including passive data collection, active data collection, and deployment-time adaptation. We provide theoretical guarantees showing that our method enhances both statistical and computational efficiency. Finally, we design practical algorithms for LLMs and conduct experiments with the `Llama-3-8B-Instruct` and `Qwen2.5-7B-Instruct` models on Ultrafeedback and Mixture2 datasets, validating the effectiveness of our approach.

## 1 Introduction

Reinforcement Learning from Human Feedback is a critical technique for training large language models using human preference feedback [Ouyang et al., 2022, Bai et al., 2022]. Typical RLHF methods involve collecting extensive data, each consisting of a prompt, a pair of responses, and a preference label indicating which response is preferred. Then, a reward model is trained to predict the human preference, and the LLM is fine-tuned based on the reward model by the RL algorithms.

Traditional RLHF methods primarily rely on fixed preference datasets, which typically suffer from limited coverage. As a result, the learned reward models struggle to generalize to out-of-distribution samples, constraining the effectiveness of the aligned models. To address this, online RLHF has emerged as a promising paradigm, enabling iterative data collection and model improvement. The general process can be described as *(i)* collect the preference data; *(ii)* update the model using the collected data. The above two steps are repeated for several iterations to boost model performance. In practice, Claude [Bai et al., 2022] and LLaMA-2 [Touvron et al., 2023] have demonstrated that online RLHF can significantly enhance model performance [Dong et al., 2024]. Theoretically, recent works [Xie et al., 2025, Cen et al., 2025] indicate that online exploration can improve the statistical

---

[*]Equal contribution.
[†]Correspondence: Peng Zhao <zhaop@lamda.nju.edu.cn>

39th Conference on Neural Information Processing Systems (NeurIPS 2025).

Table 1: Comparison between previous works and our work in terms of the statistical and computational efficiency across different online RLHF settings. The column "Context" and "Action" represent the context and action are determined by the environment (🌐) or the algorithm (🔍). For the computational efficiency (time and storage), we highlight the dependence on the $t$ at iteration $t$. Here, $d$ is the feature dimension, $T$ is the total number of iterations, $\kappa$ is the non-linearity coefficient, $\Phi = \mathbb{E}_{x \sim \rho}[\phi(x, \pi^*(x))]$ is the concentrability vector, $V_T$ and $\mathcal{H}_T$ are two local norms satisfying $\|\Phi\|_{\mathcal{H}_T^{-1}} \leq \sqrt{\kappa}\|\Phi\|_{V_T^{-1}}$ (*: amortized complexity over $T$).

| Setting | Context | Action | Gap/Regret | Time | Storage | Reference |
|---------|---------|--------|------------|------|---------|-----------|
| Passive | 🌐 | 🌐 | $\widetilde{\mathcal{O}}\big(\sqrt{d} \cdot \kappa\|\Phi\|_{V_T^{-1}}\big)$ | $\mathcal{O}(\log T)^*$ | $\mathcal{O}(t)$ | Zhu et al. [2023] |
|         |   |   | $\widetilde{\mathcal{O}}\big(\sqrt{d} \cdot \|\Phi\|_{\mathcal{H}_T^{-1}}\big)$ | $\mathcal{O}(1)$ | $\mathcal{O}(1)$ | Ours (Theorem 1) |
| Active | 🔍 | 🔍 | $\widetilde{\mathcal{O}}\big(d\sqrt{\kappa/T}\big)$ | $\mathcal{O}(t \log t)$ | $\mathcal{O}(t)$ | Das et al. [2025] |
|        |   |   | $\widetilde{\mathcal{O}}\big(d\sqrt{\kappa/T}\big)$ | $\mathcal{O}(1)$ | $\mathcal{O}(1)$ | Ours (Theorem 2) |
| Deploy | 🌐 | 🔍 | $\widetilde{\mathcal{O}}\big(d\kappa\sqrt{T}\big)$ | $\mathcal{O}(t \log t)$ | $\mathcal{O}(t)$ | Saha et al. [2023] |
|        |   |   | $\widetilde{\mathcal{O}}\big(d\sqrt{\kappa T}\big)$ | $\mathcal{O}(1)$ | $\mathcal{O}(1)$ | Ours (Theorem 3) |

efficiency of RLHF. Beyond performance gains, online RLHF serves as a crucial step toward agentic applications, where models can continuously integrate environmental feedback to enable real-time interaction, adaptive reasoning, and autonomous decision-making [Silver and Sutton, 2025].

Despite its empirical success, online RLHF may introduces significant computational challenges. Specifically, the typical process of online RLHF involves continuously integrating newly collected data into the historical dataset and re-optimizing the model from scratch over the expanded dataset. While this strategy is statistically efficient, its computational and storage costs scale linearly with the number of iterations, which becomes prohibitive in long-term iterations, especially on edge devices where computation and memory resources are inherently limited. This raises a pressing question:

*Can we design online RLHF algorithms that are both statistically and computationally efficient?*

In this work, we provide an affirmative answer to this question in the setting of contextual preference bandits with linearly parameterized reward functions. Specifically, building on recent theoretical advancements in RLHF [Zhu et al., 2023, Das et al., 2025, Ji et al., 2025], we formulate the RLHF problem as a contextual dueling bandit problem [Yue et al., 2012, Saha, 2021]. While prior work has explored this formulation, most existing methods focus on statistical efficiency and overlook the growing computational burden. To bridge this gap, inspired by recent progress in bandit learning [Zhang and Sugiyama, 2023, Li et al., 2024], we introduce a novel *one-pass* reward modeling method based on the online mirror descent framework with a tailored local norm that captures second-order information. Unlike traditional approaches, our method eliminates the need to store historical data and achieves constant-time updates per iteration, i.e., the computational cost remains invariant with respect to the cumulative number of iterations. We then apply our method to several online RLHF settings, including passive data collection, active data collection, and deployment-time adaptation. We establish theoretical guarantees showing that our method improves both statistical and computational efficiency. Table 1 summarizes the comparison of our method with the existing works.

To enable usage in LLMs, we develop practical variants of our method. Direct computation and storage of the Hessian matrix is prohibitively expensive; thus, we propose an efficient approximation using Hessian-Vector Products (HVP) combined with conjugate gradient descent, avoiding explicit second-order information and relying only on first-order computation. Additionally, we employ rejection sampling to approximate model uncertainty in a computationally efficient manner. With the above techniques, we conduct experiments using the LLaMA-3-8B-Instruct [Llama Team, 2023] and Qwen2.5-7B-Instruct [Qwen Team, 2024] models on the Ultrafeedback [Cui et al., 2024] and Mixture2 [Dong et al., 2024] datasets. Experimental results validate the effectiveness of our method.

To summarize, our contributions are as follows:

- By formulating the RLHF problem as a contextual dueling bandit, we propose a novel one-pass reward modeling algorithm and establish the corresponding estimation error bound. Our method is built upon the online mirror descent framework and incorporates a carefully designed local norm that captures second-order information for improved learning efficiency.

- We apply our method to a broad range of online RLHF settings, including passive data collection, active data collection, and deployment-time adaptation. For each setting, we design tailored algorithms and establish corresponding theoretical guarantees, demonstrating that our approach achieves improved statistical and computational efficiency compared to existing methods.
- We develop practical algorithms by approximating the update using Hessian-Vector Products combined with conjugate gradient descent, and estimating uncertainty via rejection sampling. Based on the above techniques, we conduct empirical evaluations using the `LLaMA-3-8B-Instruct` and `Qwen2.5-7B-Instruct` models on the Ultrafeedback and Mixture2 datasets, showing that our method improves both statistical and computational efficiency compared to existing methods.

**Organization.** Section 2 reviews the related work. Section 3 introduces the problem setup. Section 4 presents our proposed one-pass reward modeling method and section 5 applies it to various online RLHF settings. Section 6 provides practical versions of our method. Section 7 presents experimental results. Section 8 concludes the paper. The proofs and experiment details are deferred to the appendix.

## 2 Related Work

In this section, we review the works most closely related to ours, including online RLHF, contextual dueling bandits, and active learning.

**Online RLHF.** Traditional RLHF methods predominantly rely on fixed datasets, which often suffer from limited data coverage. Consequently, the resulting reward models struggle to generalize to out-of-distribution samples, thereby limiting the effectiveness of the aligned models. To overcome this limitation, online RLHF has emerged as a promising alternative, enabling iterative data collection and continuous model refinement. The works [Dong et al., 2023, Guo et al., 2024, Yuan et al., 2024, Wu et al., 2025] have demonstrated that online iterative variants of direct preference learning algorithms significantly outperform their offline counterparts. Xiong et al. [2024] identified key challenges in offline RLHF and theoretically demonstrated the potential benefits of online exploration. Recent work has incorporated optimism-driven bonus terms into the objective to encourage exploration in online RLHF [Xie et al., 2025, Cen et al., 2025, Zhang et al., 2025, Zhao et al., 2025]. These approaches primarily focus on the sample efficiency, but do not consider the accompanying increase in computational complexity. To improve computational efficiency, Foster et al. [2025] tackled the challenge of enumerating an exponentially large response space. Differently, our work focuses on alleviating the computational burden that scales linearly with the number of iterations in online RLHF.

**Contextual Dueling Bandits and RL.** Dueling bandits are a variant of the multi-armed bandit problem in which the learner sequentially selects a pair of arms and receives binary feedback [Yue et al., 2012]. The contextual dueling bandit framework extends this setting by incorporating contextual information [Dudík et al., 2015, Saha, 2021, Bengs et al., 2022]. Within this framework, Saha [2021] studied the $K$-armed contextual dueling bandit problem, and Saha et al. [2023] further extended it to the reinforcement learning setting. Additionally, Sekhari et al. [2023] investigated the contextual dueling bandit problem under an active learning paradigm, where the learner adaptively queries to minimize both regret and the number of queries. To move beyond linear reward functions, Verma et al. [2025a] introduced the neural dueling bandit problem, modeling the reward function using neural networks. These prior works commonly rely on maximum likelihood estimation to learn the reward function, leading to computational complexity that grows linearly with the number of iterations. In contrast, we propose algorithms that maintain constant per-iteration computational complexity.

**Active Learning.** Active learning is a paradigm that aims to reduce the labeling cost by selecting the most informative samples for annotation [Settles, 2009]. In general, existing work can be categorized into two settings: pool-based and stream-based. The pool-based setting [Seung et al., 1992, Freund et al., 1997, Huang et al., 2010] involves the learner iteratively selecting a batch of informative samples from a large unlabeled pool, querying their labels, updating the model, and repeating this process. In contrast, the stream-based setting [Cesa-Bianchi et al., 2004, 2006, Cacciarelli and Kulahci, 2024] requires the learner to sequentially observe data points and decide in real time whether to query their labels. Within the context of RLHF, Das et al. [2025] and Verma et al. [2025b] studied pool-based active learning, while Ji et al. [2025] focused on the stream-based setting. In this work, we focus on the pool-based strategy, which can be naturally extended to the stream-based scenario.

# 3 Problem Setup

Following recent advancements in RLHF [Zhu et al., 2023, Das et al., 2025, Xiong et al., 2024], we formulate RLHF as a contextual bandit problem. Specifically, we have a set of contexts $\mathcal{X}$ and a set of possible actions $\mathcal{A}$ per context. To learn with human preference feedback, the learner selects a tuple $(x, a, a')$ to present to the human, where $x \in \mathcal{X}$ is the context, $a, a' \in \mathcal{A}$ are the actions. The human then provides a binary preference feedback $y \in \{0, 1\}$, where $y = 1$ indicates that the human prefers action $a$ over action $a'$, and $y = 0$ otherwise. We study the commonly used Bradley-Terry (BT) model in preference learning [Bradley and Terry, 1952], which assumes that the human's preference is generated by a logistic function of the difference in the rewards of the two actions.

**Definition 1** (Bradley-Terry Model). Given a context $x \in \mathcal{X}$ and two actions $a, a' \in \mathcal{A}$, the probability of the human preferring action $a$ over action $a'$ is given by $\mathbb{P}[y = 1 \mid x, a, a'] = \frac{\exp(r(x,a))}{\exp(r(x,a)) + \exp(r(x,a'))}$, where $r : \mathcal{X} \times \mathcal{A} \to \mathbb{R}$ is a latent reward function.

To facilitate theoretical analysis, following prior works [Zhu et al., 2023, Cen et al., 2025], we consider the linear realizable setting, where the reward function is parameterized by a linear model.

**Assumption 1.** It holds that $r(x, a) = \phi(x, a)^\top \theta^*$ where $\phi(x, a) : \mathcal{X} \times \mathcal{A} \to \mathbb{R}^d$ is the known and fixed feature map, and $\theta^* \in \mathbb{R}^d$ is the unknown parameter vector. Furthermore, we assume $\|\phi(x, a)\|_2 \le L$ for all $x \in \mathcal{X}$ and $a \in \mathcal{A}$ and $\theta^* \in \Theta$ where $\Theta = \{\theta \in \mathbb{R}^d \mid \|\theta\|_2 \le B\}$.

**Remark 1.** While this setting is a simplification of the real-world problem, it serves as a useful and analytically tractable starting point. Specifically, the feature mapping $\phi$ can be obtained by removing the final layer of a pre-trained large language model, with $\theta^*$ corresponding to the weights of that layer. Moreover, this assumption can be further relaxed by allowing model misspecification [Jin et al., 2020] and neural function approximation [Du et al., 2024, Verma et al., 2025b].

Then, we can rewrite the probability as $\mathbb{P}[y = 1 \mid x, a, a'] = \sigma(\phi(x, a)^\top \theta^* - \phi(x, a')^\top \theta^*)$, where $\sigma(w) = \frac{1}{1 + \exp(-w)}$. Next, we introduce a key quantity that captures learning complexity.

**Definition 2.** Let $\dot{\sigma}(w) = \sigma(w)(1 - \sigma(w))$ be the derivative function of $\sigma$, the *non-linearity coefficient* $\kappa$ is defined as $\kappa = \max_{x \in \mathcal{X}, a, a' \in \mathcal{A}, \theta \in \Theta} \frac{1}{\dot{\sigma}(\phi(x,a)^\top \theta - \phi(x,a')^\top \theta)}$.

Intuitively, the quantity $\kappa$, defined as the inverse of the derivative, characterizes the learning difficulty of the reward function. In particular, a smaller derivative leads to a larger $\kappa$, implying that the model output changes less for the same input variation and thus the function is harder to learn. By direct calculation, we have $\kappa \le 3 + \exp(2BL)$. Therefore, $\kappa$ can be exceedingly large, exhibiting an exponential dependence on the magnitude of the features and the model parameters.

# 4 Our Framework

In this section, we first introduce the general framework for online RLHF. We then present our one-pass reward modeling method. Finally, we show the theoretical guarantee of our method.

## 4.1 General framework for online RLHF

The general process of online RLHF involves iteratively collecting data and updating the model based on the collected data. At iteration $t$, the process can be formulated as:

(i) **New data collection**: Sample a prompt $x_t$ and two responses $a_t$ and $a'_t$, query the oracle to obtain the preference label $y_t \in \{0, 1\}$, expand the dataset $\mathcal{D}_{t+1} = \mathcal{D}_t \cup \{(x_t, a_t, a'_t, y_t)\}$.

(ii) **Reward modeling**: Train a reward model $r_{t+1}$ using the historical dataset $\mathcal{D}_{t+1}$.

(iii) **Policy optimization (Optional)**: Update the policy $\pi_{t+1}$ using the reward model $r_{t+1}$.

A key challenge in online RLHF is that the reward model needs to be trained on the entire historical dataset at each iteration, which is computationally expensive. Specifically, let $z_t = \phi(x_t, a_t) - \phi(x_t, a'_t)$ be the feature difference, given the historical dataset $\mathcal{D}_{t+1} = \{(x_i, a_i, a'_i, y_i)\}_{i=1}^{t}$, the reward model is estimated via maximum likelihood estimation as

$$\widehat{\theta}_{t+1} = \arg\min_{\theta \in \mathbb{R}^d} \sum_{i=1}^{t} \ell_i(\theta), \text{ where } \ell_t(\theta) = -y_t \log(\sigma(z_t^\top \theta)) - (1 - y_t) \log(1 - \sigma(z_t^\top \theta)). \quad (1)$$

However, Eq. (1) does not admit a closed-form solution, requiring iterative optimization techniques, such as gradient descent, to achieve an $\varepsilon$-accurate estimate. As discussed by Faury et al. [2022], obtaining such accuracy with MLE typically requires $\mathcal{O}(\log(1/\varepsilon))$ optimization steps. Since the loss function is defined over the entire historical dataset, each iteration incurs a computational cost of $\mathcal{O}(t)$ gradient evaluations. In practice, $\varepsilon$ is often set to $1/t$ to ensure that the optimization error does not dominate the overall estimation error. As a result, the total computational complexity at iteration $t$ becomes $\mathcal{O}(t \log t)$, a cost that is prohibitive for long-term online RLHF applications.

## 4.2 One-pass reward modeling

Drawing inspiration from recent advancements in logistic bandits [Faury et al., 2022, Zhang and Sugiyama, 2023] and multinomial logit MDPs [Li et al., 2024], we propose a novel one-pass reward modeling method that reduces the complexity to constant time per iteration. First, define the gradient $g_t(\theta)$ and Hessian $H_t(\theta)$ of loss $\ell_t(\theta)$ as $g_t(\theta) = (\sigma(z_t^\top \theta) - y_t) z_t$ and $H_t(\theta) = \dot{\sigma}(z_t^\top \theta) z_t z_t^\top$.

**Implicit OMD.** To improve the computational efficiency, Faury et al. [2022] observed that the cumulative past log-loss is strongly convex and can therefore be well approximated by a quadratic function. Building on this observation, they proposed the following update rule:

$$\bar{\theta}_{t+1} = \arg\min_{\theta \in \Theta} \left\{ \ell_t(\theta) + \frac{1}{2\eta} \left\| \theta - \bar{\theta}_t \right\|_{\bar{\mathcal{H}}_t}^2 \right\}, \tag{2}$$

where $\bar{\mathcal{H}}_t = \sum_{i=1}^{t-1} H_i(\bar{\theta}_{i+1}) + \lambda I$ is the local norm, and $\eta$ is the step size. The optimization problem can be decomposed into two terms. The first term is the instantaneous log-loss $\ell_t(\theta)$, which accounts for the information of the current sample. The second consists of a quadratic proxy for the past losses constructed through the sequence $\{\bar{\theta}_i\}_{i \leq t}$. A key component is the design of the local norm $\bar{\mathcal{H}}_t$, which approximates the Hessian matrix by $H_i(\bar{\theta}_{i+1})$ at a *lookahead* point $\bar{\theta}_{i+1}$. Such a Hessian matrix effectively captures local information and is crucial for ensuring statistical efficiency.

The update rule in Eq. (2) benefits from a one-pass data processing property, which eliminates the need to store the entire historical dataset. However, the optimization problem in Eq. (2) still does not have a closed-form solution. But since the loss is defined only on the current sample, it requires only $\mathcal{O}(1)$ gradient computations per step, leading to a total computational complexity of $\mathcal{O}(\log t)$ at iteration $t$. This represents a significant improvement over the $\mathcal{O}(t \log t)$ complexity of the MLE estimator in Eq. (1). Nevertheless, the computational complexity of the implicit OMD is still increasing with the number of iterations, which motivates us to design a constant-time method.

**Standard OMD.** To enhance computational efficiency, a natural alternative is to replace this formulation with the standard OMD framework, which permits a closed-form solution and thus eliminates the need for iterative optimization. However, the standard OMD minimizes a first-order approximation of the loss function, which sacrifices several key properties compared to its implicit counterpart, as demonstrated by Campolongo and Orabona [2020]. Specifically, the standard OMD formulation updates using $g_t(\theta_t)$, whereas the implicit OMD updates the algorithm approximately with the subsequent sub-gradient, $g_t(\theta_{t+1})$. This distinction results in a notable gap in the convergence rates of the two methods. To this end, we propose to approximate the current loss $\ell_t(\theta)$ using a second-order Taylor expansion, drawing inspiration from Zhang and Sugiyama [2023]. Define the second-order approximation of $\ell_t(\theta)$ as $\widetilde{\ell}_t(\theta) = \ell_t(\widetilde{\theta}_t) + g_t(\widetilde{\theta}_t)^\top (\theta - \widetilde{\theta}_t) + \frac{1}{2} \|\theta - \widetilde{\theta}_t\|_{H_t(\widetilde{\theta}_t)}^2$. Then,

we replace the loss $\ell_t(\theta)$ in Eq. (2) with the approximation $\widetilde{\ell}_t(\theta)$, leading to the update rule:

$$\widetilde{\theta}_{t+1} = \arg\min_{\theta \in \Theta} \left\{ \langle g_t(\widetilde{\theta}_t), \theta \rangle + \frac{1}{2\eta} \|\theta - \widetilde{\theta}_t\|_{\widetilde{\mathcal{H}}_t}^2 \right\}, \tag{3}$$

where $\eta$ is the step size and $\widetilde{\mathcal{H}}_t = \mathcal{H}_t + \eta H_t(\widetilde{\theta}_t)$ is the local norm with $\mathcal{H}_t \triangleq \sum_{i=1}^{t-1} H_i(\widetilde{\theta}_{i+1}) + \lambda I$. Eq. (3) can be solved with a projected gradient step with the following equivalent form:

$$\widetilde{\theta}'_{t+1} = \widetilde{\theta}_t - \eta \widetilde{\mathcal{H}}_t^{-1} g_t(\widetilde{\theta}_t), \quad \widetilde{\theta}_{t+1} = \arg\min_{\theta \in \Theta} \|\theta - \widetilde{\theta}'_{t+1}\|_{\widetilde{\mathcal{H}}_t}^2.$$

Thus, the estimator $\widetilde{\theta}_{t+1}$ provides a closed-form solution, leading to a $\mathcal{O}(1)$ computational complexity per iteration. Furthermore, since the estimator processes the samples in a one-pass manner, it mitigates the memory burden associated with computing the gradient of the full dataset. These properties make the method particularly suitable for edge devices, where both memory and computational resources are severely constrained. The detailed process of our proposed method is presented in Algorithm 1.

| **Algorithm 1** One-Pass Reward Modeling | **Algorithm 2** Passive Data Collection |
|---|---|
| **Input:** Preference data $(x_t, a_t, a'_t, y_t)$ | **Input:** Regularization parameter $\lambda$, step size $\eta$ |
| 1: Define the loss function $\ell_t(\theta)$ as Eq. (1) | 1: Initialize $\widetilde{\theta}_1 = \mathbf{0}$ and $\widetilde{\mathcal{H}}_1 = \lambda I$ |
| 2: Update $\widetilde{\mathcal{H}}_t = \mathcal{H}_t + \eta H_t(\widetilde{\theta}_t)$ | 2: **for** $t = 1, 2, \ldots, T$ **do** |
| 3: Compute $\widetilde{\theta}'_{t+1} = \widetilde{\theta}_t - \eta \widetilde{\mathcal{H}}_t^{-1} g_t(\widetilde{\theta}_t)$ | 3:    Observe preference data $(x_t, a_t, a'_t, y_t)$ |
| 4: Compute $\widetilde{\theta}_{t+1} = \arg\min_{\theta \in \Theta} \|\theta - \widetilde{\theta}'_{t+1}\|^2_{\widetilde{\mathcal{H}}_t}$ | 4:    $\widetilde{\theta}_{t+1}$ = Algorithm 1 $(x_t, a_t, a'_t, y_t)$ |
| | 5: **end for** |
| 5: Update $\mathcal{H}_{t+1} = \mathcal{H}_t + H_t(\widetilde{\theta}_{t+1})$ | 6: Construct $\widetilde{J}_{T+1}(\pi)$ as in Eq. (4) |
| **Output:** $\widetilde{\theta}_{t+1}$ | **Output:** $\pi_{T+1} = \arg\max_{\pi \in \Pi} \widetilde{J}_{T+1}(\pi)$ |

### 4.3 Theoretical guarantee

Note that the update rule in Eq. (3) is a special case of online mirror descent, specifically:

$$\widetilde{\theta}_{t+1} = \arg\min_{\theta \in \Theta} \left\{ \eta \langle g_t(\widetilde{\theta}_t), \theta \rangle + \mathcal{D}_{\psi_t}(\theta, \widetilde{\theta}_t) \right\},$$

where $\psi_t(\theta) = \frac{1}{2}\|\theta\|^2_{\widetilde{\mathcal{H}}_t}$ is the regularizer and $\mathcal{D}_{\psi_t}(\theta, \widetilde{\theta}_t) = \psi_t(\theta) - \psi_t(\widetilde{\theta}_t) - \langle \nabla\psi_t(\widetilde{\theta}_t), \theta - \widetilde{\theta}_t \rangle$ is Bregman divergence. Leveraging the modern analysis of online mirror descent [Zhao et al., 2024, Zhang and Sugiyama, 2023], we derive the following estimation error bound.

**Lemma 1.** *Let $\delta \in (0, 1]$, set $\eta = (1/2)\log 2 + (BL + 1)$ and $\lambda = 84\sqrt{2}\eta(dL^2 + BL^3)$, define $\mathcal{C}_t = \{\theta \in \Theta \mid \|\theta - \widetilde{\theta}_t\|_{\mathcal{H}_t} \leq \widehat{\beta}_t \triangleq \mathcal{O}(\sqrt{d}\log(t/\delta))\}$. Then, we have $\Pr\left[\forall t \geqslant 1, \theta^* \in \mathcal{C}_t\right] \geqslant 1 - \delta$.*

**Comparison with MLE.** For the MLE estimator in Eq. (1), prior works [Zhu et al., 2023, Das et al., 2025, Ji et al., 2025] have shown $\|\theta - \widetilde{\theta}_t\|_{V_t} \leq \widetilde{\mathcal{O}}(\kappa\sqrt{d})$, where $V_t = \sum_{i=1}^{t-1} z_i z_i^\top + \lambda I$. By the definition of $\mathcal{H}_t$, it holds that $\mathcal{H}_t \succeq \kappa^{-1}V_t$, Lemma 1 implies that $\|\theta - \widetilde{\theta}_t\|_{V_t} \leq \sqrt{\kappa}\|\theta - \widetilde{\theta}_t\|_{\mathcal{H}_t} \leq \widetilde{\mathcal{O}}(\sqrt{\kappa}d)$. This result shows that Lemma 1 improves upon previous bounds by at least a factor of $\sqrt{\kappa}$.

## 5 Applications in Three Online RLHF Scenarios

In this section, we apply our framework to three distinct RLHF scenarios, including online RLHF with passive data collection, active data collection, and deployment-time adaptation.

### 5.1 Online RLHF with passive data collection

We first consider the passive data collection setting, where the algorithm cannot control the data collection process. At each iteration, the learner obtains $(x_t, a_t, a'_t, y_t)$ and updates by Eq. (3). We adopt the "pessimism in the face of uncertainty" principle and define the value function $\widetilde{J}_{t+1}(\pi)$ as

$$\widetilde{J}_{T+1}(\pi) = (\mathbb{E}_{x \sim \rho}[\phi(x, \pi(x))])^\top \widetilde{\theta}_{T+1} - \widetilde{\beta}_{T+1}\|\mathbb{E}_{x \sim \rho}[\phi(x, \pi(x))]\|_{\mathcal{H}_{T+1}^{-1}}. \tag{4}$$

where $\rho$ is the context distribution. The policy $\pi_{T+1}$ is selected as $\pi_{T+1} = \arg\max_{\pi \in \Pi} \widetilde{J}_{T+1}(\pi)$. The detailed procedure is present in Algorithm 2, and we show it enjoys the following guarantee.

**Theorem 1.** *Set parameters as in Lemma 1, with probability at least $1 - \delta$, Algorithm 2 ensures*

$$\mathsf{SubOpt}(\pi_{T+1}) = \mathbb{E}_{x \sim \rho}[r(x, \pi^*(x)) - r(x, \pi_{T+1}(x))] \leq \widetilde{\mathcal{O}}\left(\sqrt{d} \cdot \|\mathbb{E}_{x \sim \rho}[\phi(x, \pi^*(x))]\|_{\mathcal{H}_{T+1}^{-1}}\right),$$

*where $\rho$ is the context distribution and $\pi^*$ is the optimal policy.*

**Remark 2.** The term $\|\mathbb{E}_{x \sim \rho}[\phi(x, \pi^*(x))]\|_{\mathcal{H}_{T+1}^{-1}}$ is usually referred to "concentrability coefficient" in the literature. It measures the distribution shift between the optimal policy and the collected data.

**Remark 3.** For statistical efficiency, since $\mathcal{H}_t \succeq \kappa^{-1}V_t$, Theorem 1 improves the $\widetilde{\mathcal{O}}(\sqrt{d}\kappa \cdot \|\mathbb{E}_{x \sim \rho}[\phi(x, \pi^*(x))]\|_{V_{T+1}^{-1}})$ result of Zhu et al. [2023] at least by a factor of $\sqrt{\kappa}$. Regarding computational efficiency, their algorithm has a total storage complexity of $\mathcal{O}(T)$ and a time complexity of $\mathcal{O}(T \log T)$, leading to an amortized per-iteration cost of $\mathcal{O}(\log T)$. In contrast, our algorithm maintains a strict $\mathcal{O}(1)$ complexity per iteration, offering a substantial computational advantage.

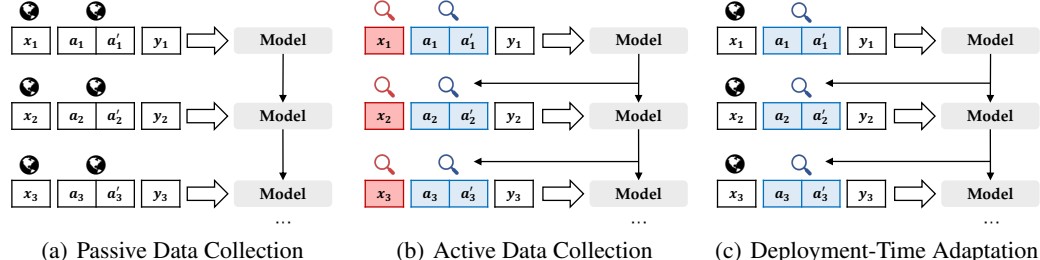

Figure 1: Different settings of online RLHF. Contexts and actions selected by the environment (🌐) are shown in grey, while those selected by the algorithm (🔍) are highlighted in color.

## 5.2 Online RLHF with active data collection

As established in Theorem 1, the sub-optimality gap depends on the concentrability coefficient, which quantifies the distributional mismatch between the optimal policy and the collected data. In this subsection, we propose an active data collection method that removes this dependency.

**Active Data Collection.** At each iteration, we select a triplet $(x_t, a_t, a_t')$ to query for human feedback $y_t$, and then update the reward model using our one-pass method as defined in Eq. (3). To guide data acquisition, we adopt an active selection strategy that queries the sample with the highest predictive uncertainty under the current reward model. Specifically, the next query is chosen by solving:

$$(x_{t+1}, a_{t+1}, a_{t+1}') = \underset{x,a,a' \in \mathcal{X} \times \mathcal{A} \times \mathcal{A}}{\arg\max} \left\{ \left\| \phi(x, a) - \phi(x, a') \right\|_{\mathcal{H}_{t+1}^{-1}} \right\}. \tag{5}$$

**Policy Optimization.** After $T$ rounds, we define the reward as the average of all the past estimations $\widetilde{r}_{T+1}(x, a) = \frac{1}{T+1} \sum_{t=1}^{T+1} \phi(x, a)^\top \widetilde{\theta}_t$. The policy is given by $\pi_{T+1}(x) = \arg\max_{a \in \mathcal{A}} \widetilde{r}_{T+1}(x, a)$.

The detailed procedure is present in Algorithm 3. We show it enjoys the following guarantee.

**Theorem 2.** *Set parameters as in Lemma 1, with probability at least $1 - \delta$, Algorithm 3 ensures*

$$\mathsf{SubOpt}(\pi_{T+1}) = \mathbb{E}_{x \sim \rho} \left[ r(x, \pi^*(x)) - r(x, \pi_{T+1}(x)) \right] \leq \widetilde{\mathcal{O}}\big(d\sqrt{\kappa/T}\big),$$

*where $\rho$ is the context distribution and $\pi^*$ is the optimal policy.*

**Remark 4.** We attain the same sub-optimality gap as Das et al. [2025], but improve the computational efficiency significantly. Our algorithm has an $\mathcal{O}(1)$ time and space complexity per round, while their MLE estimator needs $\mathcal{O}(t \log t)$ time and $\mathcal{O}(t)$ space complexity at iteration $t$.

## 5.3 Online RLHF with deployment-time adaptation

In this section, we consider the deployment-time adaptation setting, where users provide input contexts in an online manner, and the learner generates responses while simultaneously collecting feedback to improve the model. In this scenario, the learner faces a dual objective: selecting actions that maximize rewards to ensure a positive user experience, while also choosing actions that yield informative feedback to facilitate continual model improvement. To this end, we consider the measure: $\mathrm{Reg}_T = \sum_{t=1}^T \left( r\left(x_t, \pi^*(x_t)\right) - \frac{1}{2}\left(r\left(x_t, a_t\right) + r\left(x_t, a_t'\right)\right)\right)$, where $\pi^*$ is the optimal policy.

**Action selection.** At each iteration, given a prompt $x_t$ from the user, the learner selects two actions $a_t$ and $a_t'$ and obtain the feedback $y_t$. The learner must select actions that are both informative and with high rewards. To this end, we choose the first action $a_{t+1}$ to maximize the estimated reward, i.e.,

$$a_{t+1} = \underset{a \in \mathcal{A}}{\arg\max}\, \phi(x_{t+1}, a)^\top \widetilde{\theta}_{t+1}. \tag{6}$$

The second action $a_{t+1}'$ aims to maximize the reward and the distance between the two actions, i.e.,

$$a_{t+1}' = \underset{a' \in \mathcal{A}}{\arg\max} \left\{ \phi(x_{t+1}, a')^\top \widetilde{\theta}_{t+1} + \widetilde{\beta}_{t+1} \| \phi(x_{t+1}, a') - \phi(x_{t+1}, a_{t+1}) \|_{\mathcal{H}_{t+1}^{-1}} \right\}. \tag{7}$$

The overall algorithm is summarized in Algorithm 4. We show it enjoys the following regret bound.

| **Algorithm 3** Active Data Collection | **Algorithm 4** Deployment-Time Adaptation |
|---|---|
| **Input:** Regularization parameter $\lambda$, step size $\eta$ | **Input:** Regularization parameter $\lambda$, step size $\eta$ |
| 1: Initialize $\widetilde{\theta}_1 = \mathbf{0}$ and $\mathcal{H}_1 = \lambda I$ | 1: Initialize $\widetilde{\theta}_1 = 0$ and $\mathcal{H}_1 = \lambda I$. |
| 2: **for** $t = 1, 2, \ldots, T$ **do** | 2: **for** $t = 1, 2, \ldots, T$ **do** |
| 3:    Choose $(x_t, a_t, a'_t)$ as Eq. (5), observe $y_t$ | 3:    Observes the context $x_t$. |
| 4:    $\widetilde{\theta}_{t+1} = $ Algorithm 1 $(x_t, a_t, a'_t, y_t)$ | 4:    Selects $a_t$ and $a'_t$ as Eq. (6) and Eq. (7) |
| 5: **end for** | 5:    Observe the preference feedback $y_t$ |
| 6: Set $\widetilde{r}_{T+1}(x, a) = \frac{1}{T+1} \sum_{t=1}^{T+1} \phi(x, a)^\top \widetilde{\theta}_t$ | 6:    $\widetilde{\theta}_{t+1} = $ Algorithm 1 $(x_t, a_t, a'_t, y_t)$ |
| **Output:** $\pi_{T+1}(x) = \arg\max_{a \in \mathcal{A}} \widetilde{r}_{T+1}(x, a)$ | 7: **end for** |

**Theorem 3.** *For any $\delta \in (0, 1]$, set parameters as in Lemma 1, Algorithm 4 ensures the regret satisfies* $\mathrm{Reg}_T \leq \widetilde{\mathcal{O}}\big(d\sqrt{\kappa T}\big)$ *with probability at least $1 - \delta$.*

**Remark 5.** Our result improves upon Saha et al. [2023] in both computational and statistical efficiency. Statistically, Theorem 3 improves their $\widetilde{\mathcal{O}}\big(d\kappa\sqrt{T}\big)$ result by a factor of $\sqrt{\kappa}$. Computationally, our algorithm has an $\mathcal{O}(1)$ time and space complexity per round, while their MLE estimator needs $\mathcal{O}(t \log t)$ time and $\mathcal{O}(t)$ space complexity at iteration $t$ due to optimization over the historical data.

# 6 Practical Implementation

While the proposed one-pass algorithm completely removes the need to store historical data and achieves constant-time updates per iteration, its computational cost still exhibits an implicit dependence on the feature dimension $d$, which can become non-negligible in large-scale model optimization. To further alleviate this issue, we introduce in this section several empirical approximation techniques designed to reduce the effective dependence on dimensionality and enhance practical efficiency.

## 6.1 Computation of inverse Hessian

Although the OMD update in Eq. (3) enjoys one-pass property, it requires the computation of matrix inversion. Specifically, by omitting the projection operation, Eq. (3) can be rewritten as $\widetilde{\theta}_{t+1} = \widetilde{\theta}_t - \eta \widetilde{\mathcal{H}}_t^{-1} g_t(\widetilde{\theta}_t)$ where $\widetilde{\mathcal{H}}_t = \sum_{i=1}^{t-1} H_i(\widetilde{\theta}_{i+1}) + \eta H_t(\widetilde{\theta}_t) + \lambda I$. Computing the full $\widetilde{\mathcal{H}}_t^{-1}$ directly incurs a time complexity of $\mathcal{O}(d^3)$, which is prohibitive for LLMs as $d$ is typically large.

This cost can be reduced to $\mathcal{O}(d^2)$ by applying the Sherman-Morrison-Woodbury formula, leveraging the fact that the Hessian is a rank-one update. Specifically, for a matrix of the form $A + \mathbf{x}\mathbf{x}^\top$ where $A$ is invertible and $\mathbf{x}$ is a vector, the inverse is given by $(A + \mathbf{x}\mathbf{x}^\top)^{-1} = A^{-1} - \frac{A^{-1}\mathbf{x}\mathbf{x}^\top A^{-1}}{1 + \mathbf{x}^\top A^{-1}\mathbf{x}}$, requiring only $\mathcal{O}(d^2)$ time. Nevertheless, even this reduced complexity remains costly for large models.

To further reduce the computational burden to $\mathcal{O}(d)$, we employ the Hessian-vector product technique combined with conjugate gradient descent [Boyd and Vandenberghe, 2004]. Instead of explicitly computing $\widetilde{\mathcal{H}}_t^{-1}$, we define $v_t = \widetilde{\mathcal{H}}_t^{-1} g_t(\widetilde{\theta}_t)$ and solve the linear system $\widetilde{\mathcal{H}}_t v_t = g_t(\widetilde{\theta}_t)$ using the conjugate gradient method. The required matrix–vector product decomposes as $\widetilde{\mathcal{H}}_t v_t = \sum_{i=1}^{t-1} H_i(\widetilde{\theta}_{i+1}) v_t + \lambda v_t + \eta H_t(\widetilde{\theta}_t) v_t$. For the first term, materializing and storing all past Hessians $H_i(\widetilde{\theta}_{i+1})$ is infeasible. We therefore absorb their effect into the second term by replacing $\lambda$ with $\lambda_t = \lambda_0 \cdot \min\{1, f(t/T)\}$, where $f(\cdot)$ is a monotonic increasing function, such as a linear or logarithmic function. The last term can be computed via the Pearlmutter trick as $H_t(\widetilde{\theta}_t) v_t = \nabla_\theta \big(\nabla_\theta \ell_t(\theta)^\top v_t\big)\big|_{\theta = \widetilde{\theta}_t}$. Each iteration therefore requires only HVPs and vector operations, yielding an overall $\mathcal{O}(d)$ per-iteration cost with a small fixed number of iterations.

## 6.2 Computation of model uncertainty

In both online RLHF with active data collection and deployment-time adaptation, our algorithm utilizes uncertainty-driven query selection strategies. While quantifying uncertainty using the local norm induced by the inverse Hessian matrix offers strong theoretical guarantees, it is computationally

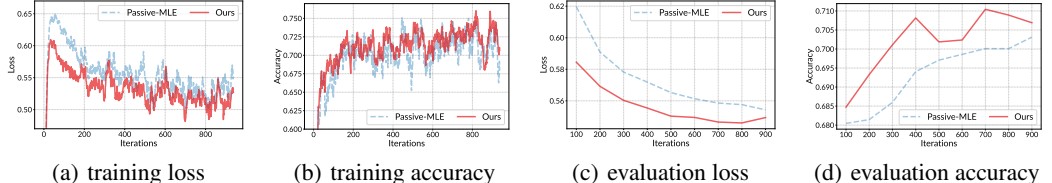

| (a) training loss | (b) training accuracy | (c) evaluation loss | (d) evaluation accuracy |

Figure 2: For online RLHF with passive data collection, we report the comparison of MLE and our method about (a) training loss, (b) training accuracy, (c) evaluation loss and (d) evaluation accuracy.

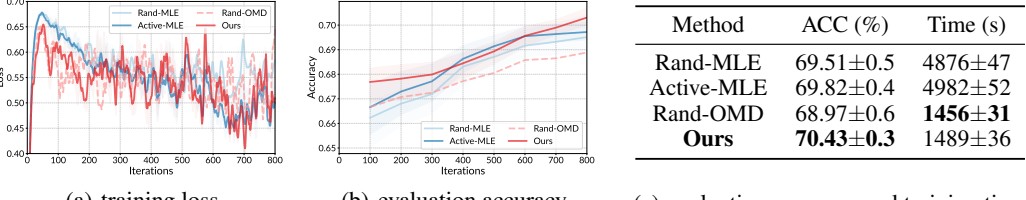

| (a) training loss | (b) evaluation accuracy | (c) evaluation accuracy and training time |

| Method | ACC (%) | Time (s) |
|---|---|---|
| Rand-MLE | 69.51±0.5 | 4876±47 |
| Active-MLE | 69.82±0.4 | 4982±52 |
| Rand-OMD | 68.97±0.6 | **1456±31** |
| **Ours** | **70.43±0.3** | 1489±36 |

Figure 3: For online RLHF with active data collection, we report the comparison of different methods about (a) training loss, (b) evaluation accuracy and (c) final evaluation accuracy and training time.

prohibitive in practice. To address this challenge, we adopt a rejection sampling-based approximation, a technique commonly employed for exploration in the RLHF literature [Nakano et al., 2021, Gulcehre et al., 2023, Dong et al., 2023, 2024]. Specifically, given a prompt, we sample $n$ independent responses by the current model, then use the trained reward function to rank the responses. Then, we use different strategies to select the response for different settings. Specifically, In active data collection, the key insight is to identify and query samples that exhibit the greatest diversity in prompt action features. To this end, we select the response with the highest predicted reward and the one with the lowest predicted reward. In deployment-time adaptation, the core idea is to select the first arm to maximize the estimated reward, while the second is chosen to balance high reward with sufficient divergence from the first. Concretely, we select the response with the highest predicted reward and another from the top-$1/q$ percentile of the reward to ensure diversity, where $q$ is a hyperparameter.

## 7 Experiments

In this section, we empirically evaluate the performance of our proposed method. [1] We first describe the experimental setup, and then present the empirical results.

### 7.1 Experiment setup

In our experiments, we employ the `Llama-3-8B-Instruct` and `Qwen2.5-7B-Instruct` as the base model for reward model. We extract features $\phi(x, a)$ using the last layer of the model, and the dimension is $d = 4096$. We use two datasets for evaluation. The first one is Ultrafeedback-binarized dataset, a pre-processed version of the original Ultrafeedback dataset [Cui et al., 2024], a widely used benchmark for RLHF. It collects about $64,000$ prompts from diverse resources, including question answering, summarization, and dialogue generation. Each data consists of a context $x$, two responses $a$ and $a'$, and a preference label $y$. We also employ a mixed dataset, Mixture2 dataset [Dong et al., 2024], which combines a variety of preference datasets, including HH-RLHF, SHP, UltraFeedback, Capybara, etc. The dataset follows the same format as the UltraFeedback-binarized dataset.

### 7.2 Experimental results

We present the experimental results for `Llama-3-8B-Instruct` on the Ultrafeedback dataset. Due to page limits, more detailed results including comparisons with Adam, DPO, full model updates, additional models of `Qwen2.5-7B-Instruct`, and Mixture2 dataset are deferred to appendix.

---

[1]The code is available at https://github.com/ZinYY/Online_RLHF

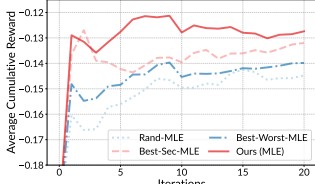
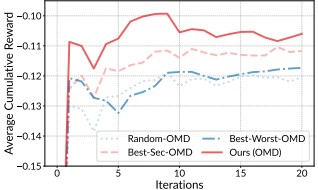
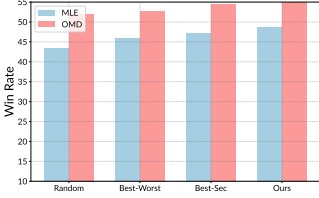

| (a) Rewards of MLE methods | (b) Rewards of OMD methods | (c) Win rates, all methods |

Figure 4: For online RLHF with deployment-time adaptation, we report (a) cumulative rewards of MLE-based methods, (b) cumulative rewards of OMD-based methods, and (c) win rates.

**Passive data collection.** We evaluate the performance of our proposed method in terms of the loss and accuracy of the reward model. We compare our OMD-based method with the MLE-based method. We randomly sample $T = 30,000$ data points from the Ultrafeedback dataset for training. Figure 2 shows the loss and accuracy vs. the number of training samples. We observe that our method converges faster to a lower loss and achieves a higher evaluation accuracy compared to baselines. The improvement is particularly pronounced in the small-sample regime ($T < 10,000$), where our method achieves a higher evaluation accuracy with the same amount of samples compared to MLE which employs conventional stochastic gradient descent (SGD) updates. This shows the superior statistical efficiency of our approach, achieving a better performance with fewer training samples.

**Active data collection.** In this setting, we only allow the algorithm to select $6,400$ samples out of the whole training datasets for training according to different selection strategies. To evaluate the effectiveness of the data selection strategy, we compare our method with the random selection strategy. We evaluate the performance of the MLE-based method and our proposed OMD-based method. Figure 3 demonstrates that our OMD-based method achieves comparable performance with the MLE-based method for both data collection strategies, while improving the training time by approximately three times. Moreover, our data selection strategy outperforms the random selection strategy, showing that our method can effectively select informative data to improve the performance.

**Deployment-time adaptation.** We divide the dataset into 20 chunks and process them sequentially to simulate the deployment scenario. We compare our action selection strategy with (i): random selection, (ii): select the best and second best actions, and (iii): select the best and worst actions. We combine the above strategies with MLE-based and OMD-based methods. We report both the average cumulative rewards and win rates of each method, where the win rate is defined as the proportion of pairwise comparisons in which a method outperforms all others. As shown in Figure 4, our action selection strategy outperforms the baselines for both MLE-based and OMD-based methods. This validates the effectiveness of our selection strategy that balances the exploitation of high-reward responses with sufficient exploration to facilitate model improvement. Besides, the win rates show that our OMD-based method achieves competitive performance with the MLE-based method.

## 8 Conclusion

In this work, we address a key challenge in online RLHF, where the computational complexity typically grows linearly with the number of iterations. To overcome this limitation, we propose a novel one-pass algorithm that eliminates the need to store historical data and achieves constant-time complexity per iteration. Our approach is built upon the online mirror descent framework with a carefully designed local norm. We apply our method to three online RLHF settings and design tailored algorithms for each scenario. We provide both theoretical guarantees and efficient implementations, demonstrating that our approach improves statistical and computational efficiency over existing methods. Finally, we validate the effectiveness of our method through extensive experiments.

While our work advances both the statistical and computational understanding of online RLHF, several important directions remain for future exploration. First, we assume a fixed feature mapping for the reward model; however, in practice, this mapping may evolve throughout the training process. Analyzing the impact of such dynamically changing feature representations presents a compelling direction for future research. Second, although our analysis is based on the Bradley-Terry model, extending the framework to other preference models, such as the Plackett-Luce model [Luce, 1959, Plackett, 1975], is another promising avenue that may broaden the applicability of our results.

## Acknowledgments

This research was supported by National Science and Technology Major Project (2022ZD0114800) and NSFC (U23A20382, 62206125). Peng Zhao was supported in part by the Xiaomi Foundation.

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

## A Useful Lemmas

**Lemma 2.** *For any $t \in [T]$, define the second-order approximation of the loss function $\ell_t(\theta)$ at the estimator $\widetilde{\theta}_t$ as*

$$\widetilde{\ell}_t(\theta) = \ell_t(\widetilde{\theta}_t) + \langle \nabla \ell_t(\widetilde{\theta}_t), \theta - \widetilde{\theta}_t \rangle + \frac{1}{2}\|\theta - \widetilde{\theta}_t\|^2_{H_t(\widetilde{\theta}_t)}.$$

*Then, for the following update rule*

$$\widetilde{\theta}_{t+1} = \underset{\theta \in \Theta}{\arg\min} \left\{ \widetilde{\ell}_t(\theta) + \frac{1}{2\eta}\|\theta - \widetilde{\theta}_t\|^2_{\mathcal{H}_t} \right\},$$

*it holds that*

$$\|\widetilde{\theta}_{t+1} - \theta^*\|^2_{\mathcal{H}_{t+1}}$$
$$\leq 2\eta \left( \sum_{i=1}^{t} \ell_i(\theta^*) - \sum_{i=1}^{t} \ell_i(\widetilde{\theta}_{i+1}) \right) + 4\lambda B^2 + 12\sqrt{2}BL^3\eta \sum_{i=1}^{t} \|\widetilde{\theta}_{i+1} - \widetilde{\theta}_i\|^2_2 - \sum_{i=1}^{t} \|\widetilde{\theta}_{i+1} - \widetilde{\theta}_i\|^2_{\mathcal{H}_i}.$$

*Proof.* Based on the analysis of (implicit) OMD update (see Lemma 5), for any $i \in [T]$, we have

$$\langle \nabla \widetilde{\ell}_i(\widetilde{\theta}_{i+1}), \widetilde{\theta}_{i+1} - \theta^* \rangle \leqslant \frac{1}{2\eta} \left( \|\widetilde{\theta}_i - \theta^*\|^2_{\mathcal{H}_i} - \|\widetilde{\theta}_{i+1} - \theta^*\|^2_{\mathcal{H}_i} - \|\widetilde{\theta}_{i+1} - \widetilde{\theta}_i\|^2_{\mathcal{H}_i} \right)$$

According to Lemma 6, we have

$$\ell_i(\widetilde{\theta}_{i+1}) - \ell_i(\theta^*) \leqslant \langle \nabla \ell_i(\widetilde{\theta}_{i+1}), \widetilde{\theta}_{i+1} - \theta^* \rangle - \frac{1}{\zeta}\|\widetilde{\theta}_{i+1} - \theta^*\|^2_{\nabla^2 \ell_i(\widetilde{\theta}_{i+1})},$$

where $\zeta = \log 2 + 2(LB + 1)$. Then, by combining the above two inequalities, we have

$$\ell_i(\widetilde{\theta}_{i+1}) - \ell_i(\theta^*) \leqslant \langle \nabla \ell_i(\widetilde{\theta}_{i+1}) - \nabla \widetilde{\ell}_i(\widetilde{\theta}_{i+1}), \widetilde{\theta}_{i+1} - \theta^* \rangle$$
$$+ \frac{1}{\zeta} \left( \|\widetilde{\theta}_i - \theta^*\|^2_{\mathcal{H}_i} - \|\widetilde{\theta}_{i+1} - \theta^*\|^2_{\mathcal{H}_{i+1}} - \|\widetilde{\theta}_{i+1} - \widetilde{\theta}_i\|^2_{\mathcal{H}_i} \right).$$

We can further bound the first term of the right-hand side as:

$$\langle \nabla \ell_i(\widetilde{\theta}_{i+1}) - \nabla \widetilde{\ell}_i(\widetilde{\theta}_{i+1}), \widetilde{\theta}_{i+1} - \theta^* \rangle = \langle \nabla \ell_i(\widetilde{\theta}_{i+1}) - \nabla \ell_i(\widetilde{\theta}_i) - \nabla^2 \ell_i(\widetilde{\theta}_i)(\widetilde{\theta}_{i+1} - \widetilde{\theta}_i), \widetilde{\theta}_{i+1} - \theta^* \rangle$$
$$= \langle D^3 \ell_i(\xi_{i+1})[\widetilde{\theta}_{i+1} - \widetilde{\theta}_i](\widetilde{\theta}_{i+1} - \widetilde{\theta}_i), \widetilde{\theta}_{i+1} - \theta^* \rangle$$
$$\leqslant 3\sqrt{2}L\|\widetilde{\theta}_{i+1} - \theta^*\|_2 \|\widetilde{\theta}_{i+1} - \widetilde{\theta}_i\|^2_{\nabla^2 \ell_i(\xi_{i+1})}$$
$$\leqslant 6\sqrt{2}BL\|\widetilde{\theta}_{i+1} - \widetilde{\theta}_i\|^2_{\nabla^2 \ell_i(\xi_{i+1})}$$
$$\leqslant 6\sqrt{2}BL^3\|\widetilde{\theta}_{i+1} - \widetilde{\theta}_i\|^2_2,$$

where the second equality holds by the mean value theorem, the first inequality holds by the self-concordant-like property of $\ell_i(\cdot)$ in Lemma 3, and the last inequality holds by $\widetilde{\theta}_{i+1}$ and $\theta^*$ belong to $\Theta = \{\theta \in \mathbb{R}^d, \|\theta\|_2 \leq B\}$, and $\nabla^2 \ell_i(\xi_{i+1}) \preceq L^2 I_d$.

Then, by taking the summation over $i$ and rearranging the terms, we obtain

$$\|\widetilde{\theta}_{t+1} - \theta^*\|^2_{\mathcal{H}_{t+1}}$$
$$\leqslant \zeta \sum_{i=1}^{t} \left( \ell_i(\theta^*) - \ell_i(\widetilde{\theta}_{i+1}) \right) + \|\widetilde{\theta}_1 - \theta^*\|^2_{\mathcal{H}_1} + 6\sqrt{2}BL^3\zeta \sum_{i=1}^{t} \|\widetilde{\theta}_{i+1} - \widetilde{\theta}_i\|^2_2 - \sum_{i=1}^{t} \|\widetilde{\theta}_{i+1} - \widetilde{\theta}_i\|^2_{\mathcal{H}_i}$$
$$\leqslant \zeta \sum_{i=1}^{t} \left( \ell_i(\theta^*) - \ell_i(\widetilde{\theta}_{i+1}) \right) + 4\lambda B^2 + 6\sqrt{2}BL^3\zeta \sum_{i=1}^{t} \|\widetilde{\theta}_{i+1} - \widetilde{\theta}_i\|^2_2 - \sum_{i=1}^{t} \|\widetilde{\theta}_{i+1} - \widetilde{\theta}_i\|^2_{\mathcal{H}_i},$$

where the last inequality is by $\|\widetilde{\theta}_1 - \theta^*\|^2_{\mathcal{H}_1} \leq \lambda \|\widetilde{\theta}_1 - \theta^*\|^2_2 \leq 4\lambda B^2$. Set $\zeta = 2\eta$ ends the proof. ∎

# B Proof of Lemma 1

*Proof.* Based on Lemma 2, we have

$$\|\widetilde{\theta}_{t+1} - \theta^*\|^2_{\mathcal{H}_{t+1}}$$

$$\leq 2\eta\left(\sum_{i=1}^{t}\ell_i(\theta^*) - \sum_{i=1}^{t}\ell_i(\widetilde{\theta}_{i+1})\right) + 4\lambda B^2 + 12\sqrt{2}BL^3\eta\sum_{i=1}^{t}\|\widetilde{\theta}_{i+1} - \widetilde{\theta}_i\|_2^2 - \sum_{i=1}^{t}\|\widetilde{\theta}_{i+1} - \widetilde{\theta}_i\|^2_{\mathcal{H}_i}.$$

It remains to bound the right-hand side of the above inequality in the following. The most challenging part is to bound the term $\sum_{i=1}^{t}\ell_i(\theta^*) - \sum_{i=1}^{t}\ell_i(\widetilde{\theta}_{i+1})$. This term might seem straightforward to control, as it can be observed that $\theta^* = \arg\min_{\theta\in\mathbb{R}^d}\bar{\ell}(\theta) \triangleq \mathbb{E}_{y_i}[\ell_i(\theta)]$, where $\ell_i(\theta)$ serves as an empirical observation of $\bar{\ell}(\theta)$. Consequently, the loss gap term seemingly can be bounded using appropriate concentration results. However, a caveat lies in the fact that the update of the estimator $\widetilde{\theta}_{i+1}$ depends on $\ell_i$, or more precisely $y_i$, making it difficult to directly apply such concentrations.

To address this issue, following the analysis in Zhang and Sugiyama [2023], we decompose the loss gap into two components by introducing an intermediate term. Specifically, we define the softmax function as

$$[\sigma_i(q)]_1 = \frac{\exp(q)}{1 + \exp(q)} \quad \text{and} \quad [\sigma_i(q)]_0 = \frac{1}{1 + \exp(q)},$$

where $[\cdot]_i$ denotes the $i$-th element of the vector. Then, the loss function $\ell_i(\theta)$ can be rewritten as

$$\ell(q_t, y_t) = -\mathbb{1}_{\{y_t=1\}} \cdot \log\left([\sigma(q_t)]_1\right) - \mathbb{1}_{\{y_t=0\}} \cdot \log\left([\sigma(q_t)]_0\right).$$

Then, we define the pseudo-inverse function of $\sigma^{-1}(p)$ with

$$[\sigma^{-1}(p)]_1 = \log(q/(1-q)) \quad \text{and} \quad [\sigma^{-1}(p)]_0 = \log((1-p)/p).$$

Then, we decompose the regret into two terms by introducing an intermediate term.

$$\sum_{i=1}^{t}\ell_i(\theta^*) - \sum_{i=1}^{t}\ell_i(\widetilde{\theta}_{i+1}) = \underbrace{\sum_{i=1}^{t}\ell_i(\theta^*) - \sum_{i=1}^{t}\ell_i(q_i, y_i)}_{\texttt{term (a)}} + \underbrace{\sum_{i=1}^{t}\ell_i(q_i, y_i) - \sum_{i=1}^{t}\ell_i(\widetilde{\theta}_{i+1})}_{\texttt{term (b)}}$$

where $q_i$ is an aggregating forecaster for logistic loss defined by $q_i = \sigma^{-1}(\mathbb{E}_{\theta\sim P_i}[\sigma(\theta^\top z_i)])$ and $P_i = \mathcal{N}(\widetilde{\theta}_i, (1 + c\mathcal{H}_i^{-1}))$ is the Gaussian distribution with mean $\widetilde{\theta}_i$ and covariance $(1 + c\mathcal{H}_i^{-1})$, where $c > 0$ is a constant to be specified later. It remains to bound the terms $\texttt{term (a)}$ and $\texttt{term (b)}$, which were initially analyzed in Zhang and Sugiyama [2023] and further refined by Lee and Oh [2024]. Specifically, using Lemmas F.2 and F.3 in Lee and Oh [2024], we can bound them as follows.

For $\texttt{term (a)}$, let $\delta \in (0, 1)$ and $\lambda \geq 1$. With probability at least $1 - \delta$, for all $t \in [T]$, we have

$$\texttt{term (a)} \leq 11 \cdot (3\log(1 + 2t) + 2 + LB)\log\left(\frac{2\sqrt{1+2t}}{\delta}\right) + 2.$$

For $\texttt{term (b)}$, let $\lambda \geq \max\{2, 72cd\}$. Then, for all $t \in [T]$, we have

$$\texttt{term (b)} \leq \frac{1}{2c}\sum_{i=1}^{t}\left\|\widetilde{\theta}_{i+1} - \widetilde{\theta}_i\right\|^2_{\mathcal{H}_i} + \sqrt{6}cd\log\left(1 + \frac{2tB^2}{d\lambda}\right)$$

Combing the above two bounds, we have

$$\left\|\widetilde{\theta}_{t+1} - \theta^*\right\|^2_{\mathcal{H}_{t+1}} \leq 12\sqrt{2}BL^3\eta\sum_{i=1}^{t}\left\|\widetilde{\theta}_{i+1} - \widetilde{\theta}_i\right\|_2^2 + \left(\frac{\eta}{c} - 1\right)\sum_{i=1}^{t}\left\|\widetilde{\theta}_{i+1} - \widetilde{\theta}_i\right\|^2_{\mathcal{H}_i} + C.$$

where $C = 22\eta(3\log(1 + 2t) + 2 + LB)\log\left(\frac{2\sqrt{1+2t}}{\delta}\right) + 4\eta + 2\eta\sqrt{6}cd\log\left(1 + \frac{2tL^2}{d\lambda}\right) + 4\lambda B^2$. Setting $c = 7\eta/6$ and $\lambda \geq 84\sqrt{2}BL^3\eta$, we have

$$12\sqrt{2}BL^3\eta\sum_{i=1}^{t}\left\|\widetilde{\theta}_{i+1} - \widetilde{\theta}_i\right\|_2^2 + \left(\frac{\eta}{c} - 1\right)\sum_{i=1}^{t}\left\|\widetilde{\theta}_{i+1} - \widetilde{\theta}_i\right\|^2_{\mathcal{H}_i}$$

$$\leq \left(12\sqrt{2}BL^3\eta - \frac{\lambda}{7}\right) \sum_{i=1}^{t} \left\|\widetilde{\theta}_{i+1} - \widetilde{\theta}_i\right\|_2^2$$

$$\leq 0.$$

Note that $84\sqrt{2}\left(BL^3 + dL^2\right)\eta \geq \max\left\{2L^2, 72cdL^2, 84\sqrt{2}BL^3\eta\right\}$, so we set $\lambda \geq 84\sqrt{2}\left(BL^3 + dL^2\right)\eta$. As we have $\eta = (1/2)\log 2 + (BL + 1)$, we have

$$\left\|\widetilde{\theta}_{t+1} - \theta^*\right\|_{\mathcal{H}_{t+1}} \leq \mathcal{O}\left(\sqrt{d}\log(t/\delta)\right).$$

This finishes the proof. ∎

## C  Proof of Theorem 1

*Proof.* Define $J(\pi) = \mathbb{E}_{x\sim\rho}[r\left(x, \pi(x)\right)]$, we have

$$\mathsf{SubOpt}\left(\pi_T\right) = \left(J\left(\pi^*\right) - \widetilde{J}\left(\pi^*\right)\right) + \left(\widetilde{J}\left(\pi^*\right) - \widetilde{J}\left(\pi_T\right)\right) + \left(\widetilde{J}\left(\pi_T\right) - J\left(\pi_T\right)\right).$$

Since $\pi_T$ is the optimal policy under expected value $\widetilde{J}(\pi)$, i.e., $\widetilde{J}(\pi_T) = \max_{\pi\in\Pi}\widetilde{J}(\pi)$, we have

$$\widetilde{J}\left(\pi^*\right) - \widetilde{J}\left(\pi_T\right) \leq 0 \tag{8}$$

For the third term, we have with probability at least $1 - \delta$, it holds that

$$\widetilde{J}\left(\pi_T\right) - J\left(\pi_T\right) = \min_{\theta\in\mathcal{C}_T}\mathbb{E}_{x\sim\rho}\left[\theta^\top\phi(s, \pi_T(s))\right] - \mathbb{E}_{x\sim\rho}\left[\theta^{*\top}\phi(s, \pi_T(s))\right] \leq 0, \tag{9}$$

where the last inequality holds by $\theta^* \in \mathcal{C}_T$ with probability at least $1 - \delta$.

For the first term, we have with probability at least $1 - \delta$, it holds that

$$\begin{aligned}
J\left(\pi^*\right) &- \widetilde{J}\left(\pi^*\right) \\
&= \mathbb{E}_{x\sim\rho}\left[(\theta^*)^\top\phi(s, \pi^*(s))\right] - \min_{\theta\in\mathcal{C}_T}\mathbb{E}_{x\sim\rho}\left[\theta^\top\phi(s, \pi^*(s))\right] \\
&= \sup_{\theta\in\mathcal{C}_T}\mathbb{E}_{x\sim\rho}\left[\left(\theta^* - \widetilde{\theta}_T + \widetilde{\theta}_T - \theta\right)^\top\phi(x, \pi^*(x))\right] \\
&= \mathbb{E}_{x\sim\rho}\left[\left(\theta^* - \widetilde{\theta}_T\right)^\top\phi(x, \pi^*(x))\right] + \sup_{\theta\in\mathcal{C}_T}\mathbb{E}_{x\sim\rho}\left[\left(\widetilde{\theta}_T - \theta\right)^\top\phi(x, \pi^*(x))\right] \\
&\leq \left(\|\theta^* - \widetilde{\theta}_T\|_{\mathcal{H}_T} + \sup_{\theta\in\mathcal{C}_T}\|\theta - \widetilde{\theta}_T\|_{\mathcal{H}_T}\right) \cdot \left\|\mathbb{E}_{x\sim\rho}[\phi(x, \pi^*(x))]\right\|_{\mathcal{H}_T^{-1}},
\end{aligned}$$

where the first inequality holds by the Cauchy-Schwarz inequality.

Since it holds $\theta^* \in \mathcal{C}_T$ with probability at least $1 - \delta$ by Lemma 1, we have $\|\theta^* - \widetilde{\theta}_T\|_{\mathcal{H}_T} \leq \widetilde{\beta}_T$ and $\sup_{\theta\in\mathcal{C}_T}\|\theta - \widetilde{\theta}_T\|_{\mathcal{H}_T} \leq \widetilde{\beta}_T$. Thus, we obtain

$$J\left(\pi^*\right) - \widetilde{J}\left(\pi^*\right) \leq 2\widetilde{\beta}_T \cdot \left\|\mathbb{E}_{x\sim\rho}[\phi(x, \pi^*(x))]\right\|_{\mathcal{H}_T^{-1}}. \tag{10}$$

Combining Eq. (8), Eq. (9), and Eq. (10) and substituting $\widetilde{\beta}_T = \mathcal{O}(\sqrt{d}(\log(T/\delta))^2)$, we have with probability at least $1 - \delta$, it holds that

$$\begin{aligned}
\mathsf{SubOpt}\left(\pi_T\right) &\leq 2\widetilde{\beta}_T \cdot \left\|\mathbb{E}_{x\sim\rho}[\phi(x, \pi^*(x))]\right\|_{\mathcal{H}_T^{-1}} \\
&\leq \mathcal{O}\left(\sqrt{d}\left(\log\frac{T}{\delta}\right)^2 \cdot \left\|\mathbb{E}_{x\sim\rho}[\phi(x, \pi^*(x))]\right\|_{\mathcal{H}_T^{-1}}\right).
\end{aligned}$$

This completes the proof. ∎

# D  Proof of Theorem 2

*Proof.* Let the sub-optimality gap for a context $x \in \mathcal{X}$ be denoted as $\mathsf{SubOpt}(x)$. Thus, for any $\delta \in (0, 1)$, with probability at least $1 - \delta$, we have

$$\mathsf{SubOpt}(x) = \left(\phi\left(x, \pi^*(x)\right) - \phi\left(x, \pi_T(x)\right)\right)^\top \theta^*$$

$$\leq \left(\phi\left(x, \pi^*(x)\right) - \phi\left(x, \pi_T(x)\right)\right)^\top \theta^* + \left(\phi\left(x, \pi_T(x)\right) - \phi\left(x, \pi^*(x)\right)\right)^\top \left(\frac{1}{T} \sum_{t=1}^T \widetilde{\theta}_t\right)$$

$$= \left(\phi\left(x, \pi^*(x)\right) - \phi\left(x, \pi_T(x)\right)\right)^\top \left(\theta^* - \frac{1}{T} \sum_{t=1}^T \widetilde{\theta}_t\right)$$

$$= \frac{1}{T} \sum_{t=1}^T \left(\phi\left(x, \pi^*(x)\right) - \phi\left(x, \pi_T(x)\right)\right)^\top \left(\theta^* - \widetilde{\theta}_t\right)$$

$$\leq \frac{1}{T} \sum_{t=1}^T \|\phi\left(x, \pi^*(x)\right) - \phi\left(x, \pi_T(x)\right)\|_{\mathcal{H}_t^{-1}} \left\|\theta^* - \widetilde{\theta}_t\right\|_{\mathcal{H}_t}$$

$$\leq \frac{\widetilde{\beta}_T}{T} \sum_{t=1}^T \|\phi\left(x, \pi^*(x)\right) - \phi\left(x, \pi_T(x)\right)\|_{\mathcal{H}_t^{-1}},$$

where the first inequality is due to the fact that $\left(\phi\left(x, \pi_T(x)\right) - \phi\left(x, \pi^*(x)\right)\right)^\top \left(\frac{1}{T} \sum_{t=1}^T \widetilde{\theta}_t\right) \geq 0$ by the design of $\pi_T(x)$, the second is due to the Cauchy-Schwarz inequality, and the last inequality is due to $\|\theta^* - \widetilde{\theta}_t\|_{\mathcal{H}_t} \leq \beta_T$ with probability at least $1 - \delta$ by Lemma 1.

By our algorithm's choice $(x_t, a_t, a_t') = \arg\max_{x \in \mathcal{X}, a, a' \in \mathcal{A}} \|\phi(x, a) - \phi(x, a')\|_{\mathcal{H}_t^{-1}}$, we have

$$\sum_{t=1}^T \|\phi\left(x, \pi^*(x)\right) - \phi\left(x, \pi_T(x)\right)\|_{\mathcal{H}_t^{-1}} \leq \sum_{t=1}^T \|\phi\left(x_t, a_t\right) - \phi\left(x_t, a_t'\right)\|_{\mathcal{H}_t^{-1}} = \sum_{t=1}^T \|z_t\|_{\mathcal{H}_t^{-1}}.$$

Furthermore, by the definition of $\mathcal{H}_t$, we have

$$\mathcal{H}_t = \lambda I_d + \sum_{s=1}^{t-1} \dot{\sigma}\left(z_s^\top \widetilde{\theta}_{s+1}\right) z_s z_s^\top \geq \lambda I_d + \frac{1}{\kappa} \sum_{s=1}^{t-1} z_s z_s^\top = \frac{1}{\kappa}\left(\kappa \lambda I_d + \sum_{s=1}^{t-1} z_s z_s^\top\right) = \frac{1}{\kappa} V_t.$$

Thus, we have

$$\sum_{t=1}^T \|z_t\|_{\mathcal{H}_t^{-1}} \leq \sqrt{\kappa} \sum_{t=1}^T \|z_t\|_{V_t^{-1}} \leq \sqrt{\kappa} \sqrt{T \sum_{t=1}^T \|z_t\|_{V_t^{-1}}^2} \leq \sqrt{2\kappa d T \log\left(1 + \frac{4TL^2}{\lambda \kappa d}\right)},$$

where the first inequality holds by the fact that $\mathcal{H}_t \succeq \frac{1}{\kappa} V_t$, the second inequality holds by the Cauchy-Schwarz inequality, and the last inequality holds by the elliptic potential lemma in Lemma 4. Thus, we have for any context $x \in \mathcal{X}$,

$$\mathsf{SubOpt}(x) \leq \frac{\widetilde{\beta}_T}{T} \sqrt{2\kappa d T \log\left(1 + \frac{4TL^2}{\lambda \kappa d}\right)}.$$

By the definition of $\mathsf{SubOpt}(\pi_T)$, we have with probability at least $1 - \delta$,

$$\mathsf{SubOpt}\left(\pi_T\right) = \mathbb{E}_{x \sim \rho}\left[\mathsf{SubOpt}(x)\right] \leq \frac{\widetilde{\beta}_T}{T} \sqrt{2\kappa d T \log\left(1 + \frac{T}{\lambda \kappa d}\right)} \leq \widetilde{\mathcal{O}}\left(d\sqrt{\frac{\kappa}{T}}\right).$$

This finishes the proof. ∎

# E    Proof of Theorem 3

*Proof.* We first analyze the instantaneous regret at round $t$. For any $\delta \in (0, 1)$, with probability at least $1 - \delta$, it holds that

$$
\begin{aligned}
&\big(r(x_t, \pi^*(x_t)) - r(x_t, a_t)\big) + \big(r(x_t, \pi^*(x_t)) - r(x_t, a_t')\big) \\
&= \big(\phi(x_t, \pi^*(x_t)) - \phi(x_t, a_t)\big)^\top \theta^* + \big(\phi(x_t, \pi^*(x_t)) - \phi(x_t, a_t')\big)^\top \theta^* \\
&= 2\big(\phi(x_t, \pi^*(x_t)) - \phi(x_t, a_t)\big)^\top \theta^* + \big(\phi(x_t, a_t) - \phi(x_t, a_t')\big)^\top \theta^* \\
&= 2\big(\phi(x_t, \pi^*(x_t)) - \phi(x_t, a_t)\big)^\top (\theta^* - \widetilde{\theta}_t) + 2\big(\phi(x_t, \pi^*(x_t)) - \phi(x_t, a_t)\big)^\top \widetilde{\theta}_t \\
&\quad + \big(\phi(x_t, a_t) - \phi(x_t, a_t')\big)^\top (\theta^* - \widetilde{\theta}_t) + \big(\phi(x_t, a_t) - \phi(x_t, a_t')\big)^\top \widetilde{\theta}_t \\
&\le 2\big\|\phi(x_t, \pi^*(x_t)) - \phi(x_t, a_t)\big\|_{\mathcal{H}_t^{-1}} \big\|\theta^* - \widetilde{\theta}_t\big\|_{\mathcal{H}_t} + \big(\phi(x_t, \pi^*(x_t)) - \phi(x_t, a_t)\big)^\top \widetilde{\theta}_t \\
&\quad + \big\|\phi(x_t, a_t) - \phi(x_t, a_t')\big\|_{\mathcal{H}_t^{-1}} \big\|\theta^* - \widetilde{\theta}_t\big\|_{\mathcal{H}_t} + \big(\phi(x_t, a_t) - \phi(x_t, a_t')\big)^\top \widetilde{\theta}_t \\
&\le 2\widetilde{\beta}_t \big\|\phi(x_t, \pi^*(x_t)) - \phi(x_t, a_t)\big\|_{\mathcal{H}_t^{-1}} + \big(\phi(x_t, \pi^*(x_t)) - \phi(x_t, a_t')\big)^\top \widetilde{\theta}_t \\
&\quad + \widetilde{\beta}_t \big\|\phi(x_t, a_t) - \phi(x_t, a_t')\big\|_{\mathcal{H}_t^{-1}} \\
&\le 2\widetilde{\beta}_t \big\|\phi(x_t, a_t') - \phi(x_t, a_t)\big\|_{\mathcal{H}_t^{-1}} + (\phi(x_t, a_t') - \phi(x_t, \pi^*(x_t)))^\top \widetilde{\theta}_t \\
&\quad + \big(\phi(x_t, \pi^*(x_t)) - \phi(x_t, a_t')\big)^\top \widetilde{\theta}_t + \widetilde{\beta}_t \big\|\phi(x_t, a_t) - \phi(x_t, a_t')\big\|_{\mathcal{H}_t^{-1}} \\
&= 3\widetilde{\beta}_t \big\|\phi(x_t, a_t) - \phi(x_t, a_t')\big\|_{\mathcal{H}_t^{-1}},
\end{aligned}
$$

where the first inequality holds by the Holder's inequality and the arm selection strategy of $a_t$ such that $\phi(x_t, \pi^*(x_t))^\top \widetilde{\theta}_t \le \phi(x_t, a_t)^\top \widetilde{\theta}_t$, the second inequality holds by $\widetilde{\theta}_t \in \mathcal{C}_t$ with probability at least $1 - \delta$ by Lemma 1, the third inequality holds by arm selection strategy of $a_t'$ such that $a_t' = \arg\max_{a \in \mathcal{A}} \phi(x_t, a)^\top \widetilde{\theta}_t + 2\widetilde{\beta} \|\phi(x_t, a) - \phi(x_t, a_t)\|_{\mathcal{H}_t^{-1}}$. Thus, we have

$$
\begin{aligned}
\mathrm{Reg}_T &= \frac{1}{2} \sum_{t=1}^T \Big( \big(r(x_t, \pi^*(x_t)) - r(x_t, a_t)\big) + \big(r(x_t, \pi^*(x_t)) - r(x_t, a_t')\big) \Big) \\
&\le \frac{3}{2} \widetilde{\beta}_T \sum_{t=1}^T \big\|\phi(x_t, a_t) - \phi(x_t, a_t')\big\|_{\mathcal{H}_t^{-1}}.
\end{aligned}
$$

By the definition of $\mathcal{H}_t$, we have

$$
\mathcal{H}_t = \lambda I_d + \sum_{s=1}^{t-1} \dot{\sigma}\big(z_s^\top \widetilde{\theta}_{s+1}\big) z_s z_s^\top \ge \lambda I_d + \frac{1}{\kappa} \sum_{s=1}^{t-1} z_s z_s^\top = \frac{1}{\kappa}\left(\kappa \lambda I_d + \sum_{s=1}^{t-1} z_s z_s^\top\right) = \frac{1}{\kappa} V_t.
$$

Thus, we have

$$
\sum_{t=1}^T \|z_t\|_{\mathcal{H}_t^{-1}} \le \sqrt{\kappa} \sum_{t=1}^T \|z_t\|_{V_t^{-1}} \le \sqrt{\kappa} \sqrt{T \sum_{t=1}^T \|z_t\|_{V_t^{-1}}^2} \le \sqrt{2\kappa dT \log\left(1 + \frac{4TL^2}{\lambda \kappa d}\right)},
$$

where the first inequality holds by the fact that $\mathcal{H}_t \succeq \frac{1}{\kappa} V_t$, the second inequality holds by the Cauchy-Schwarz inequality, and the last inequality holds by the elliptic potential lemma in Lemma 4.

Therefore, we have

$$
\mathrm{Reg}_T \le \frac{3}{2} \widetilde{\beta}_T \sqrt{2\kappa dT \log\left(1 + \frac{4\kappa TL^2}{\lambda d}\right)} \le \widetilde{\mathcal{O}}\big(d\sqrt{\kappa T}\big).
$$

where the This completes the proof. ∎

# F    Supporting Lemmas

**Definition 3** (Tran-Dinh et al. [2015])**.** A convex function $f \in \mathcal{C}^3(\mathbb{R}^m)$ is $M$-self-concordant-like function if

$$|\psi'''(s)| \leqslant M\|\mathbf{b}\|_2 \psi''(s),$$

for $s \in \mathbb{R}$ and $M > 0$, where $\psi(s) := f(\mathbf{a} + s\mathbf{b})$ for any $\mathbf{a}, \mathbf{b} \in \mathbb{R}^m$.

**Lemma 3** (Lee and Oh [2024, Proposition C.1])**.** *The loss $\ell_t(\theta)$ defined in Eq. (1) is $3\sqrt{2}L$-self-concordant-like for $\forall t \in [T]$.*

**Lemma 4** (Abbasi-Yadkori et al. [2011, Lemma 11])**.** *Suppose $x_1, \ldots, x_t \in \mathbb{R}^d$ and for any $1 \leq s \leq t$, $\|x_s\|_2 \leq L$. Let $V_t = \lambda I_d + \sum_{s=1}^{t-1} x_s x_s^\top$ for $\lambda \geq 0$. Then, we have*

$$\sum_{s=1}^t \|z_s\|_{V_s^{-1}}^2 \leq 2d \log\left(1 + \frac{tL^2}{\lambda d}\right).$$

**Lemma 5** (Campolongo and Orabona [2020, Proposition 4.1])**.** *Define $\mathbf{w}_{t+1}$ as the solution of*

$$\mathbf{w}_{t+1} = \arg\min_{\mathbf{w} \in \mathcal{V}} \left\{ \eta\ell_t(\mathbf{w}) + \mathcal{D}_\psi(\mathbf{w}, \mathbf{w}_t) \right\},$$

*where $\mathcal{V} \subseteq \mathcal{W} \subseteq \mathbb{R}^d$ is a non-empty convex set. Further supposing $\psi(\mathbf{w})$ is 1-strongly convex w.r.t. a certain norm $\|\cdot\|$ in $\mathcal{W}$, then there exists a $\mathbf{g}_t' \in \partial\ell_t(\mathbf{w}_{t+1})$ such that*

$$\langle \eta_t \mathbf{g}_t', \mathbf{w}_{t+1} - \mathbf{u} \rangle \leq \langle \nabla\psi(\mathbf{w}_t) - \nabla\psi(\mathbf{w}_{t+1}), \mathbf{w}_{t+1} - \mathbf{u} \rangle$$

*for any $\mathbf{u} \in \mathcal{W}$.*

**Lemma 6** (Zhang and Sugiyama [2023, Lemma 1])**.** *Let $\ell(\mathbf{z}, y) = \sum_{k=0}^K \mathbf{1}\{y = k\} \cdot \log\left(\frac{1}{[\sigma(\mathbf{z})]_k}\right)$ where $\sigma(\mathbf{z})_k = \frac{e^{z_k}}{\sum_{j=0}^K e^{z_j}}$, $\mathbf{a} \in [-C, C]^K$, $y \in \{0\} \cup [K]$ and $\mathbf{b} \in \mathbb{R}^K$ where $C > 0$. Then, we have*

$$\ell(\mathbf{a}, y) \geq \ell(\mathbf{b}, y) + \nabla\ell(\mathbf{b}, y)^\top(\mathbf{a} - \mathbf{b}) + \frac{1}{\log(K+1) + 2(C+1)}(\mathbf{a} - \mathbf{b})^\top \nabla^2\ell(\mathbf{b}, y)(\mathbf{a} - \mathbf{b}).$$

# G    Details of Experiments

In this section, we provide the omitted details of the experiment details and additional results.

## G.1    Implementation Details

**Datasets.**    We use the UltraFeedback-binarized dataset [Rafailov et al., 2023] for the experiments. This dataset is derived from the original UltraFeedback dataset, which comprises 64, 000 prompts sourced from diverse datasets including UltraChat, ShareGPT, Evol-Instruct, TruthfulQA, FalseQA, and FLAN. For each prompt, four model completions were generated using various open-source and proprietary language models, with GPT-4 providing comprehensive evaluations across multiple criteria including helpfulness, honesty, and truthfulness. The binarized version was constructed by selecting the completion with the highest overall score as the "chosen" response and randomly selecting one of the remaining completions as the "rejected" response, creating clear preference pairs suitable for reward modeling and direct preference optimization. This dataset structure aligns well with our experimental setup, providing a robust foundation for evaluating different preference learning approaches. The dataset's diverse prompt sources and evaluation criteria make it particularly valuable for training and evaluating reward models in a real-world context. To further tailor the dataset to our experimental setup, we organize the dataset as follows:

- Passive data collection: We randomly choose 30, 000 samples from the UltraFeedback-binarized dataset's `train_prefs` split for training. Each sample consists of a prompt and two responses with a label indicating the preferred response. We use the `test_prefs` split for evaluation.

- Active data collection: We allow the method to actively select 6,400 samples from the `train_prefs` split according to different selection strategies. The global batch size is set to 8 for training. The selection is performed iteratively, where in each iteration, the method selects the most informative samples based on its selection criterion.

---

**Algorithm 5** Efficient Update using Hessian-Vector Product with Conjugate Gradient

---

**Input:** Current parameter $\widetilde{\theta}_t$, gradient $g_t(\widetilde{\theta}_t)$, learning rate $\eta$, max CG steps $K$, parameter $\lambda_0, \epsilon$

1: Initialize $v_0 = 0$, $r_0 = g_t(\widetilde{\theta}_t)$, $p_0 = r_0$
2: Compute damping $\lambda_t = \lambda_0 \cdot \min\{1, f(t/T)\}$
3: **for** $k = 0, 1, \ldots, K - 1$ **do**
4:     Compute HVP: $\widetilde{\mathcal{H}}_t p_k = \nabla_\theta (\nabla_\theta \mathcal{L}(\theta)^\top p_k)|_{\theta = \widetilde{\theta}_t} + \lambda_t p_k$
5:     $\alpha_k = \frac{r_k^\top r_k}{p_k^\top \widetilde{\mathcal{H}}_t p_k}$, $v_{k+1} = v_k + \alpha_k p_k$, $r_{k+1} = r_k - \alpha_k \widetilde{\mathcal{H}}_t p_k$,
6:     $\beta_{k+1} = \frac{r_{k+1}^\top r_{k+1}}{r_k^\top r_k}$, $p_{k+1} = r_{k+1} + \beta_{k+1} p_k$
7:     **if** $\|r_{k+1}\| \leq \epsilon$ **then**
8:         **break**
9:     **end if**
10: **end for**
11: Update parameter: $\widetilde{\theta}_{t+1} = \widetilde{\theta}_t - \eta v_K$
**Output:** Updated parameter $\widetilde{\theta}_{t+1}$

---

- Deployment-time adaption: We use a pre-processed online variant of the UltraFeedback-binarized dataset from the `test_gen` split. The dataset is divided into 20 sequential chunks to simulate an online deployment scenario. For each chunk, we generate responses using the current policy (the foundation model of policy model is chosen to be `meta-llama / Llama-3.2-1B`), evaluate them using both the learned reward model and an oracle reward model. We choose `NCSOFT/Llama-3-OffsetBias-RM-8B` [Park et al., 2024] as the oracle reward model. After each chunk, we use the policy model to randomly generate 64 responses using different seeds. We then apply various strategies (*Random*, *Best-Two*, etc.) to select responses and construct new preference pairs, which are then used to update the reward model and the policy model.

**Update details.** As described in Section 6.1, we can implement the OMD update using the HVP with conjugate gradient descent. The full algorithm is summarized in Algorithm 5. In our experiments, we set $K = 3$ and $\lambda_0 = 0.8$ and choose the linear function $f(t/T) = t/T$ as the damping function.

### G.2 Validating the Magnitude of $\kappa$

We validate the magnitude of $\kappa$ by computing its value during the training process. The results show that $\kappa = 171.62 \pm 85.49$ during our training process, which is relatively large.

### G.3 Combined with Adam Optimizer

In previous experiments, we used SGD to update model parameters. In this section, we integrate the methods with the *Adam optimizer* [Kingma and Ba, 2015], i.e., adding the first and second momentum terms to the model updates. The results, shown in Figure 5, indicate that the Adam optimizer further enhances the performance of our method by leveraging the momentum term to accelerate convergence. With the momentum term, our method remains superior to the MLE-based method; however, the performance gap is reduced. This may be because the Adam optimizer incorporates second-order information for optimization, diminishing the advantage of our method compared to the SGD cases.

### G.4 Comparison with DPO

We also compare with DPO [Rafailov et al., 2023] in the deployment stage. As a reward-free method, DPO optimizes the policy directly using preference feedback without explicit reward modeling. To ensure a fair comparison, we initialize the policy with 400 samples and use the same dataset settings as PPO to iteratively update the policy model using the DPO algorithm. The results are illustrated in Figure 6. While DPO outperforms the random baseline (Rand-MLE), it achieves lower cumulative rewards than the methods using our action selection. This result suggests that DPO's online learning capability remains a challenge. In contrast, the reward model learned by our selection strategy effectively learned streaming data and continuously updates the policy as new data arrive, indicating that a reward model with PPO may be a more suitable choice for sequentially learning from new data.

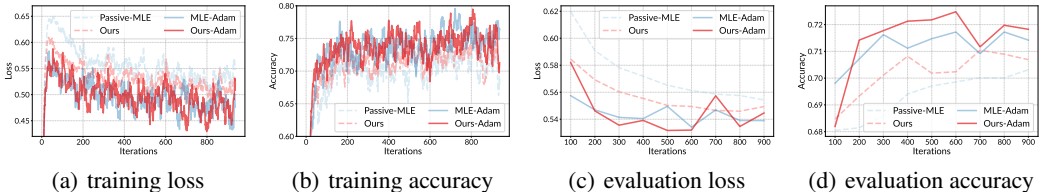

| (a) training loss | (b) training accuracy | (c) evaluation loss | (d) evaluation accuracy |

Figure 5: For online RLHF with *passive data collection*, we compare our proposed method and MLE [Zhu et al., 2023] in with passive data collection combined with *Adam*. We report the average accuracy and loss of the reward model during the training process.

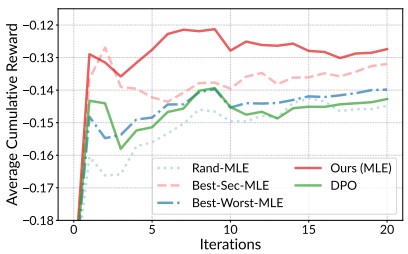

Figure 6: Comparison of DPO and our method in deployment-time adaptation.

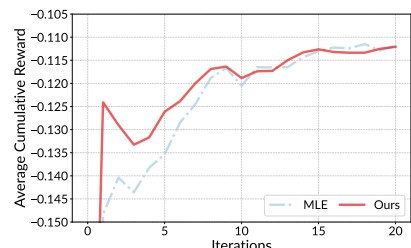

Figure 7: Comparison of different methods for full update in deployment-time adaptation.

## G.5 Full Update of Reward Model

Figure 7 shows deployment-time adaptation results using the *Llama-3.2-1B* model, where we update all parameters of the reward model instead of only the final layer. Both our method and MLE use the same action selection strategy. Our approach achieves comparable performance with MLE, indicating that our OMD-based update method is still compatible with full-model updates.

## G.6 More Foundation Models and Datasets

In this section, we provide more experimental results about other foundation models and datasets.

Figure 8 shows the training and evaluation curves for reward model learning under passive data collection using the `Qwen2.5-7B-Instruct` model. We compare our method with MLE and report the loss and accuracy over training. Our method consistently shows stable training dynamics and competitive evaluation performance compared to MLE, suggesting its effectiveness in offline settings.

Figure 9 present results for online RLHF with active data collection using the same Qwen model. Figure 9(a) shows training loss curves, while Figure 9(b) reports evaluation accuracy over training iterations. Table 9(c) further compares various methods (Rand-MLE, Active-MLE, Rand-OMD, and our approach) in terms of final accuracy and training time. While Active-MLE achieves slightly higher accuracy, our method provides significant speedup in training time with comparable performance, highlighting the efficiency of our approach.

Figure 10 illustrates the deployment-time performance of various methods on the Ultrafeedback dataset. We split the dataset into 20 chunks and measure cumulative rewards across these chunks. Our method demonstrates robust adaptation capabilities, achieving competitive reward accumulation.

Finally, Figure 11 shows results on the `Llama-3-8B-Instruct` model trained on the *Mixture2* dataset in a passive data collection setup. Similar to earlier observations, our method achieves competitive or superior performance compared to MLE, both in terms of training and evaluation loss/accuracy, demonstrating its generality across different model and dataset combinations.

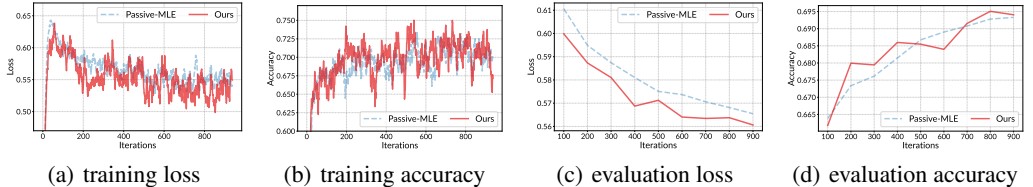

| (a) training loss | (b) training accuracy | (c) evaluation loss | (d) evaluation accuracy |

Figure 8: For `Qwen2.5-7B-Instruct` model with *passive data collection*, we compare our method with MLE. We report average accuracy and loss curve of the reward model.

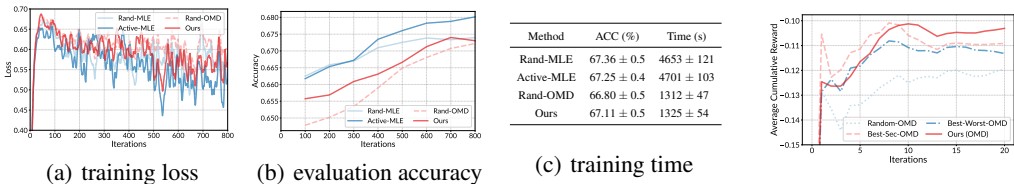

| (a) training loss | (b) evaluation accuracy | (c) training time |

Figure 9: For `Qwen2.5-7B-Instruct` with active data collection, we report the comparison of different methods about (a) training loss, (b) evaluation accuracy and (c) evaluation accuracy and training time.

Figure 10: Results of deployment-time adaptation for `Qwen2.5-7B-Instruct`.

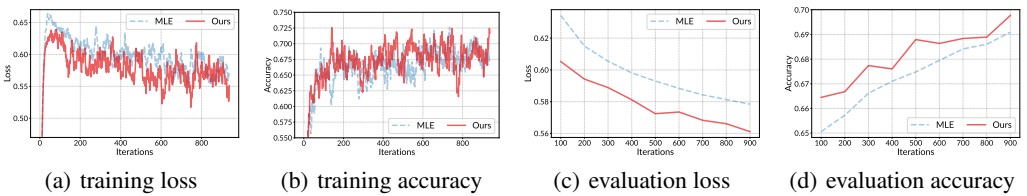

| (a) training loss | (b) training accuracy | (c) evaluation loss | (d) evaluation accuracy |

Figure 11: For online RLHF with *passive data collection* on the `Llama-3-8B-Instruct` model on the *Mixture2* dataset, we compare our method with MLE. We report average accuracy and loss curve of the reward model.

## H   Broader Impact

Our work advances the efficiency of RLHF, a central technique in aligning large language models with human values and preferences. By proposing a new one-pass reward modeling method that eliminates the need to store historical data and re-train from scratch, we reduce the computational and environmental costs commonly associated with online RLHF pipelines. This could enable the development and deployment of aligned language models by institutions with limited resources.

However, the broader deployment of RLHF, particularly in an online and adaptive setting, raises important ethical and societal considerations. On the positive side, it can enable more responsive and value-aligned AI systems, with potential applications in education, healthcare, and accessibility. Yet, the ability to iteratively adapt to user feedback in deployment may also increase the risk of reinforcing harmful biases or being gamed by adversarial users, especially in high-stakes or open-ended domains.

