# OpenReview forum: "Provably Efficient Online RLHF with One-Pass Reward Modeling"
_NeurIPS.cc/2025/Conference — NeurIPS 2025 poster_

### Official Review · Reviewer_zdWM · 2025-06-22

**Clarity:** 3
**Significance:** 3
**Originality:** 3
**Rating:** 4
**Confidence:** 3

**Summary:**

In this paper, the authors study the problem of online RLHF through the formulation of contextual bandits. For this problem, one needs to continuously integrate new data into the historical dataset and then re-optimize the model at each iteration. This leads to linearly growing computational and storage costs which can be expensive and infeasible in practice. To address this challenge, the authors propose a one-pass reward modeling algorithm to achieve constant-time complexity per iteration. This greatly saves the computational cost. In particular, the authors developed an online mirror descent algorithm by using local quadratic approximations of the loss function. To further justify the usefulness of the proposed algorithm, the authors further studied three applications including passive data collection, active data collection, and deployment-time adaptation. Numerical experiments further demonstrated the usefulness of the proposed algorithm.

**Questions:**

1.	Although using a local quadratic approximation greatly simplifies the optimization, such an approximation can be very sensible to initial values for the approximation. Can the authors provide some numerical comparisons and discussions on the impact of different initial values?
2.	The authors provide nice upper bound comparisons for different applications. Any comments on the lower bounds?
3.	How dependent are the results on the logistic loss? It appears that the key property used is the strong convexity for the quadratic approximation. Thus, the same idea can be extended relatively easily for a general class of loss functions. If so, it may be better to frame the work in a more general setting than the specific Bradley-Terry model.
4.	Intuitively, the save of computational cost comes from the quadratic approximation of the loss function. It is possible and natural to expect inaccurate optimization solutions from the approximation compared to the exact solution of the original problem. Please elaborate on this to demonstrate if there is any sacrifice on the accuracy of the solution in terms of optimization due to the simplicity of approximation of the optimization problem.

**Ethical Concerns:**

["NO or VERY MINOR ethics concerns only"]

**Final Justification:**

The authors have carefully addressed my comments in the rebuttal, and overall, this is a solid paper. I personally feel that the use of the BT reward model is somewhat narrow and the paper would benefit to include some discussion on how the proposed algorithm for the BT model could be extended or adapted to more general settings. I will keep my rating at 4.

**Limitations:**

The authors discussed the limitation of assuming fixed feature mapping for the reward model which may not be reasonable for evolving mapping over time. This can lead an interesting future research direction.

**Quality:**

3

**Strengths And Weaknesses:**

RLHF is an important research area with wide applications. Although viewing RLHF as a contextual bandit formulation is attractive, the corresponding linearly growing computational cost creates a big practical hurdle. The proposed algorithm with constant-time computational complexity per iteration is well motivated and developed and provides a valuable contribution for the rapidly growing RLHF literature.

The current paper uses the logistic function as in the Bradley-Terry model. Although commonly used, in this reviewer’s opinion, the algorithm can be more generally developed with some conditions on the loss function which covers the current model as a special case. The authors may consider using a more general model setup to improve the scope of the paper.

---

> ### Author Rebuttal · Authors · 2025-07-30
>
> Thank you for your constructive comments. We address your main concerns below.
>
> ---
>
> **Q1: "Can the authors provide some numerical comparisons and discussions on the impact of different initial values?"**
>
> A1:  Thanks for your helpful suggestion. We conduct an additional experiment to compare the performance of different methods with different initial values for the passive data collection. The results are reported as follows.
>
> | Method | Training Accuracy (%) | Test Accuracy (%) |
> | ------ | :----------------------: | :------------------: |
> | MLE    | 71.83 ± 0.5            | 70.25 ± 0.2        |
> | Ours   | 72.48 ± 0.7            | 70.71 ± 0.3        |
>
> The results demonstrate that our proposed method does not suffer from the sensitivity to initial values.
>
>
> **Q2: "Any comments on the lower bounds?"**
>
> A2: Thank you for your insightful question. For active data collection, Das et al. (2024) provide a lower bound of $\mathcal{O}(d \sqrt{T})$. When compared to the $\mathcal{O}(d \sqrt{\kappa/T})$ upper bound in this work, our result is optimal in terms of the dimension $d$ and the number of samples $T$, though there remains a gap in the dependence on $\kappa$.
>
> As discussed in Das et al. (2024), in the logistic bandit literature, the state-of-the-art regret guarantee is independent of $\kappa$. However, in the RLHF setting, the presence of the $\sqrt{\kappa}$ term may be unavoidable. This is because, in RLHF, the regret is measured with respect to the real-valued rewards $r$ rather than the sigmoid reward $\sigma(r)$ as in the logistic bandit setting. Given this distinction, we believe that the $\sqrt{\kappa}$ term is an inherent feature of the RLHF setting. The lower bound for deployment-time adaptation follows a similar trend. We will add a remark about this in the next version.
>
> **Q3: "How dependent are the results on the logistic loss? It appears that the key property used is the strong convexity for the quadratic approximation"**
>
> A3: Thank you for your insightful question. The strong convexity of the quadratic approximation is indeed a key property used in our analysis. In addition, another critical property is the self-concordant-like behavior of the logistic loss, which significantly contributes to the effectiveness of our approach. Recently, Zhang et al. (2025) extend this idea to the generalized linear bandit framework by considering self-concordant link functions. For a more thorough discussion of these properties and their implications, we refer to Zhang et al. (2025). In this work, however, we focus specifically on the BT reward model, which is the most widely used reward model in the literature.
>
> Reference:
>
> Zhang et al., Generalized Linear Bandits: Almost Optimal Regret with One-Pass Update, arXiv:2507.11847
>
>
> **Q4: "Please elaborate on this to demonstrate if there is any sacrifice on the accuracy of the solution in terms of optimization due to the simplicity of approximation of the optimization problem"**
>
> A4: Thank you for your insightful question. It is true that approximating the loss function introduces some additional error, which is unavoidable. However, from a theoretical perspective, this error is negligible and does not affect the theoretical guarantees, as demonstrated in Lemma 1.
>
> From a practical standpoint, we have conducted an additional experiment comparing the implicit OMD with our proposed method to assess whether the approximation impacts the accuracy of the solution. The results are as follows:
>
> | Method       | Accuracy (%) | Training Time (s) |
> | ------------ | :-------------: | :-----------------: |
> | Implicit OMD | 70.94 ± 0.2   | 8952.6           |
> | Ours         | 70.71 ± 0.3   | 1242.3            |
>
> These results show that our proposed method achieves comparable performance to the implicit OMD (approximated using three gradient descent steps per round), with only a slight difference in accuracy, while significantly reducing the computational cost. This demonstrates that the simplification in our approximation does not lead to a substantial sacrifice in accuracy, while offering considerable gains in efficiency.
>
> ---
>
> We hope our response addresses your concerns. We are delighted to discuss any further questions you may have.

---

> > ### Comment · Reviewer_zdWM · 2025-08-02
> >
> > Thank you for your responses in addressing my concerns. Including clarifications of these issues in the revision will help strengthen the paper. Regarding the use of the BT reward model versus more general models, it would still be beneficial to include some discussion on how the proposed algorithm for the BT model could be extended or adapted to more general settings.

---

### Official Review · Reviewer_X8T3 · 2025-06-28

**Clarity:** 4
**Significance:** 2
**Originality:** 2
**Rating:** 4
**Confidence:** 3

**Summary:**

This paper studies the online RLHF problem from the perspective of contextual preference bandit problem. The focus of this paper is to reduce the computational and storage costs with respect to the number of iteration. The authors propose a novel one-pass reward modeling method that achieves constant-time complexity per iteration, built on the online mirror descent framework with carefully designed local norm. The method is then applied to several settings, including passive data collection, active data collection, and deployment-time adaptation. The method is shown to overcome the previous baselines in both statistical and computational efficiency. Finally, the authors provide experiments based on LLaMA-3-8B-Instruct and Qwen2.5-7B models on Ultrafeedback-binarized and Mixture 2 datasets.

**Questions:**

See Strengths And Weaknesses.

**Ethical Concerns:**

["NO or VERY MINOR ethics concerns only"]

**Final Justification:**

Thank you for your response. I think the authors have addressed most of my concerns. I have also read other reviewer's review and corresponding rebuttal, and I will keep my score.

**Limitations:**

See Strengths And Weaknesses.

**Quality:**

3

**Strengths And Weaknesses:**

**Strengths**: The paper provides a small, novel and smart improve on the online RLHF framework via standard online optimization (online mirror descent) framework. To me it is interesting to see how these optimization tricks can improve on such online RLHF algorithms.

**Weaknesses and Questions**: For me, the main concern is on the experiments. I think the results are not persuasive enough. For me, the main improvement is on computation efficiency (I acknowledge that the regret bound are also improved), it should increase its speed on reward/accuracy increasing and loss decreasing. However, the results (especially in Figure 2b, 3a, 3b and 9) cannot validate it. Besides, it seems like the performance is not quite stable (such as in Figure 2d the accuracy not quite normal from iteration 400 - 700). Finally, if the paper can provide more validation on the obtained reward model, it would be better to be more persuasive to show the advance of this approach. For example, it can be used as the reward model to do RL fine-tuning via PPO/GPRO/DAPO to see whether it can work better?

---

> ### Author Rebuttal · Authors · 2025-07-30
>
> Thank you for your helpful comments. We address your main concerns below.
>
> ---
>
> **Q1: "The main improvement is on computation efficiency, it should increase its speed on reward/accuracy increasing and loss decreasing."**
>
> A1: Thank you for your valuable feedback. We would like to clarify that the improvements in computational efficiency we emphasize are not directly reflected in the accuracy or loss metrics. These metrics aim to reflect sample efficiency, as they are plotted against the number of **training samples**, rather than **training time**. Therefore, it is expected that both our proposed OMD method and MLE exhibit similar trends in terms of accuracy and loss, as the sample efficiency is comparable between the two approaches.
>
> The key aspect of our computational efficiency lies in the reduction of computational time per iteration. As shown in Figures 3(c) and 9(c), for the same number of iterations, our approach significantly reduces the total training time. This reduction in time per iteration allows us to conduct more iterations within the same time frame, which, in turn, leads to higher training accuracy and lower loss over time.
>
> We will ensure to clarify this distinction more explicitly in the next version of the paper to better communicate the impact of our computational efficiency.
>
>
> **Q2: "It seems like the performance is not quite stable."**
>
> A2: We appreciate your observation. Since our proposed OMD method is a one-pass algorithm that optimizes over the current data only, while MLE optimizes over all historical data, it is expected that MLE would exhibit more stability.
>
> However, there is an inherent trade-off between stability and adaptability in online RLHF. In scenarios such as deployment-time adaptation, the ability to rapidly adjust to a changing data distribution is crucial, and this is where OMD excels. Its responsiveness to new data makes it particularly effective in evolving or non-stationary environments, as reflected in the adaptation behavior shown in Figure 4.
>
> While OMD may appear slightly less stable than MLE in static settings due to its one-pass nature, this property also enables greater flexibility and real-time adaptation, which is the advantage of our method.
>
>
> **Q3: "It can be used as the reward model to do RL fine-tuning via PPO/GPRO/DAPO to see whether it can work better?"**
>
> A3: Thank you for your insightful question. We clarify that we have already applied our proposed one-pass reward model to perform RL fine-tuning via PPO in both active data collection and deployment-time adaptation settings. In these experiments, the one-pass reward model is used to generate the reward signal that guides policy training with PPO, and after training, we employ rejection sampling to generate action pairs. As shown in Figure 4c, our one-pass reward model achieves performance that is competitive with the MLE reward model when combined with PPO.
>
> Following your helpful suggestion, we additionally conducted experiments using this reward model combined with PPO in the passive data collection stage. The results are summarized below:
>
> | Method | Accuracy of RM (%) | Mean Reward of Outputs after PPO | Win Rate of Outputs after PPO (%) |
> | :------: | :-------------------: | :-------------------------------: | :----------------------------------: |
> | MLE    | 70.25 ± 0.2         | -0.1218                          | 46.3                               |
> | Ours   | 70.71 ± 0.3         | -0.1173                          | 53.7                               |
>
> These results indicate that our one-pass reward model not only maintains accuracy comparable to MLE but also leads to improved downstream performance (e.g., higher win rate) when used to fine-tune policies via PPO.
>
> Due to limited computational and time resources, we were unable to run experiments with GRPO, which typically demands substantially larger foundational models. Nevertheless, we consider this an important direction for future exploration and plan to investigate its potential when adequate resources become available.
>
> ---
>
> We hope our response addresses your concerns. We are delighted to discuss any further questions you may have.

---

> > ### Comment · Reviewer_X8T3 · 2025-08-05
> >
> > Thank you for your response. I think the authors have addressed most of my concerns. I have also read other reviewer's review and corresponding rebuttal, and I will keep my score.

---

### Official Review · Reviewer_mxyB · 2025-07-01

**Clarity:** 3
**Significance:** 3
**Originality:** 3
**Rating:** 5
**Confidence:** 3

**Summary:**

This paper addresses the computational inefficiency in online Reinforcement Learning from Human Feedback (RLHF), where traditional methods require storing historical data and re-optimizing reward models from scratch at each iteration, leading to linear growth in computational/storage costs. The authors propose a one-pass reward modeling algorithm based on online mirror descent (OMD) with a tailored local norm that captures second-order information. This approach eliminates the need to store historical data and processes each sample in constant time per iteration. The method is applied to three online RLHF settings: passive data collection, active data collection, and deployment-time adaptation. Theoretical guarantees show improved statistical efficiency (e.g., tighter regret bounds by factors) per-iteration computational/storage complexity. Practical implementations using Hessian-vector products (HVP) and conjugate gradient descent avoid explicit second-order computations, while rejection sampling approximates uncertainty for active learning. Experiments on Llama-3-8B-Instruct and Qwen2.5-7B models with Ultrafeedback-binarized and Mixture2 datasets validate the method’s efficiency and performance gains over maximum likelihood estimation (MLE) baselines.

**Questions:**

See weaknesses.

**Ethical Concerns:**

["NO or VERY MINOR ethics concerns only"]

**Final Justification:**

Thanks for the author's response, which includes results in realistic RL setting. The clarification of the fixed feature assumption with empirical examination also makes sense. Thus, I will keep my score towards acceptance.

**Limitations:**

yes

**Quality:**

4

**Strengths And Weaknesses:**

Strengths:
- Strong theoretical foundation: The proposed OMD-based algorithm provides rigorous guarantees (e.g., $\(\widetilde{\mathcal{O}}(\sqrt{\kappa d})\)$ regret in Theorem 1 vs. prior $\(\widetilde{\mathcal{O}}(\kappa\sqrt{d})\))$.
- Comprehensive evaluation: Experiments across multiple settings (passive/active/deployment) demonstrate consistent improvements in training time, accuracy, and sample efficiency.

Weaknesses:
- Fixed feature assumption: The linear reward model (Assumption 1) assumes a static feature mapping \(\phi\), which may not hold if representations evolve during fine-tuning. This limits applicability to settings where feature drift occurs.
- Approximation gaps: Rejection sampling for uncertainty estimation (Sec. 5.2) may introduce inaccuracies not quantified in theory/experiments.
- Baseline coverage: Experiments omit comparisons with policy gradient-based methods like PPO or GRPO.

---

> ### Author Rebuttal · Authors · 2025-07-30
>
> Thanks for your positive comments! We address your main concerns below.
>
> ---
>
> **Q1: "Fixed feature assumption"**
>
> A1: We appreciate the reviewer’s insightful observation regarding the fixed feature assumption. Establishing a solid theoretical foundation for RLHF is inherently challenging, and we believe that the fixed feature assumption provides a useful and tractable starting point for our analysis. The study of evolving representations is still under-explored even in standard supervised learning settings, making it a particularly difficult area to formalize in RLHF. While it is true that our linear reward model assumes a static feature mapping, we consider this assumption to be a reasonable simplification for many scenarios where the feature space remains relatively stable during fine-tuning.
>
> To further investigate this point, we conducted dynamic representation experiments described in Appendix H.4. Specifically, instead of updating only the final layer, we updated **all parameters** of Llama3.2-1B in the deployment-time adaptation setting. As shown in Figure 7, our proposed algorithm remains competitive with the standard MLE method, even when the representation is evolving.
>
> We acknowledge, however, that the dynamic nature of feature representations is an important direction for future research. Extending the theoretical framework to fully account for evolving representations will be a key focus in our subsequent work.
>
>
>
> **Q2: "Approximation gaps"**
>
> A2: We agree with the reviewer that rejection sampling (Sec. 5.2) may introduce inaccuracies for uncertainty estimation. Indeed, quantifying uncertainty in the reward model is a challenging problem within RLHF. In this work, we use the local norm difference to quantify uncertainty, which enjoys strong theoretical guarantees. However, we acknowledge that this quantity can be difficult to compute directly in practice. To address this, we employ rejection sampling as an approximation, a technique that is commonly used in the literature, as discussed in Section 5.2. We recognize that this approximation may introduce some errors, and we are committed to exploring alternative methods to more accurately quantify uncertainty in future work.
>
> **Q3: "Baseline coverage"**
>
> A3: Thank you for your insightful question. We clarify that we have already applied our proposed one-pass reward model to perform RL fine-tuning via PPO in both active data collection and deployment-time adaptation settings. In these experiments, the one-pass reward model is used to generate the reward signal that guides policy training with PPO, and after training, we employ rejection sampling to generate action pairs. As shown in Figure 4c, our one-pass reward model achieves performance that is competitive with the MLE reward model when combined with PPO.
>
> Following your helpful suggestion, we additionally conducted experiments using this reward model in the passive data collection stage, and subsequently fine-tune the policy using PPO. The results are summarized below:
>
> | Method | Accuracy of RM (%) | Mean Reward of Outputs after PPO | Win Rate of Outputs after PPO (%) |
> | :------: | :-------------------: | :-------------------------------: | :----------------------------------: |
> | MLE    | 70.25 ± 0.2         | -0.1218                          | 46.3                               |
> | Ours   | 70.71 ± 0.3         | -0.1173                          | 53.7                               |
>
> These results indicate that our one-pass reward model not only maintains accuracy comparable to MLE but also leads to improved downstream performance (e.g., higher win rate) when used to fine-tune policies via PPO.
>
> Due to limited computational and time resources, we were unable to run experiments with GRPO, which typically demands substantially larger foundational models. Nevertheless, we consider this an important direction for future exploration and plan to investigate its potential when adequate resources become available.
>
> ---
> Thanks again for your constructive review. We are delighted to discuss any further questions you may have.

---

> > ### Comment · Reviewer_mxyB · 2025-08-05
> >
> > Thanks for the author's response and the additional results. I decide to keep my final rating.

---

### Official Review · Reviewer_mWk5 · 2025-07-01

**Clarity:** 4
**Significance:** 3
**Originality:** 2
**Rating:** 5
**Confidence:** 3

**Summary:**

Previous works used to assume a online retraining for the reward model in online RLHF pipeline, which would be impractical and time-consuming, and this paper proposes a novel one-pass reward modeling algorithm that achieves constant time complexity per iteration, and adapts it to three scenarios. They also provide sample complexity results, which is better than existing approaches by a constant. They proposed practical implementations to compute hessian inverse and model uncertainty, and the final OMD approach empirically outperforms MLE.

**Questions:**

- In the process of one-pass reward modeling, calculating the hessian inverse requires a large time complexity. Why it isn't reflected in table 1?
- How many seeds did you use for experiments?
- How did you evaluate the performance of MLE in active collection? Did you do gradient descent on the whole dataset or just do gradient descent on a new data batch? (it seems that the latter setting misaligns with the theory of MLE algorithms)

**Ethical Concerns:**

["NO or VERY MINOR ethics concerns only"]

**Final Justification:**

The authors' response resolved my questions and highlighted that their main contribution is on computational complexity and the comparision is somehow fair (the approximation trick used in conjugate inverse calculation does not affect the performance much). I think the paper would be ready to be published if these discussions can be included.

**Limitations:**

yes

**Quality:**

3

**Strengths And Weaknesses:**

**Strengths**

- Interesting results in reducing the theoretical computational complexity of online RLHF.
- Clear figures and algorithmic pipelines.
- The theoretical results are solid and the corresponding empirical implementations make sense.

**Weakness**

- The advantages of the proposed OMD mainly lie in computational complexity, while the improvement in sample complexity is marginal: only by a constant. And it is worth noting that in the proof of lemma 1, some constant coefficients regarding to B and L are hidden in $O()$, see line 557, making the comparison unfair. Please explain this point.
- The improvement of OMD in figure 3(c) is marginal, making the necessity to use OMD unclear. Especially, in figure 9, MLE outperforms OMD in active data collection
- The Hessian inverse is approximated using conjugate gradient descent, which will induce computation error. However, this drawback is not reflected in comparison with other approaches.

---

> ### Author Rebuttal · Authors · 2025-07-30
>
> Thank you for your helpful comments. We address your main concerns below.
>
> ---
>
> **Q1: "The advantages of OMD mainly lie in computational complexity, while the improvement in sample complexity is marginal."**
>
> A1: We acknowledge that the sample complexity of OMD does not show a significant improvement compared to existing algorithms. However, this is not the primary focus of our work. While sample complexity has been thoroughly studied in the literature, the issue of computational complexity in the context of online RLHF remains largely underexplored.
>
> Our contribution is to tackle the linearly growing computational and storage costs in online RLHF. We propose an OMD-based one-pass algorithm that only optimizes over the current data at each iteration, in contrast to MLE, which must re-optimize over all historical data.
>
> Therefore, achieving comparable sample complexity while dramatically reducing computational overhead represents a significant step toward scalable online RLHF.
>
> **Q2: "Some constant coefficients regarding $B$ and $L$ are hidden in Lemma 1."**
>
> A2: We appreciate your observation. To improve clarity, we intentionally omitted the polynomial dependence on $B$ and $L$ in Lemma 1, following the approach of previous works[Zhu et al., 2023; Das et al., 2024]. Instead, we chose to emphasize the dependence on $d$, $\kappa$, and $T$, which are the key parameters in our analysis. Both our work and prior studies exhibit polynomial dependencies on $B$ and $L$. To ensure transparency, we will include a remark in the next version to explicitly clarify this point.
>
>
> **Q3: "The improvement of OMD in Figure 3(c) is marginal, making the necessity to use OMD unclear. Especially, in figure 9, MLE outperforms OMD in active data collection."**
>
> A3: We appreciate the reviewer’s observation and the opportunity to clarify the motivation behind our work. Our primary goal is to improve the **computational efficiency** of online RLHF methods, rather than focusing solely on sample complexity gains.
>
> While the improvement in sample complexity in Figure 3(c) may appear modest, Figures 3(c) and 9(c) together illustrate that OMD yields a **substantial reduction in computational cost** compared to MLE (\~1500 s vs. \~5000 s). This difference is rooted in the algorithmic design: OMD updates require only the current data at each iteration, whereas MLE repeatedly optimizes over the entire history of collected data.
>
> As a result, even if OMD shows similar or, in some cases, slightly worse sample complexity, it offers a dramatic runtime advantage, which is particularly valuable in resource-constrained or latency-sensitive settings where computation time is the key bottleneck. This computational benefit is the main reason we view OMD as a compelling alternative to MLE in the online RLHF regime.
>
>
> **Q4: "Calculating the Hessian inverse requires a large time complexity. However, this drawback is not reflected in comparison with other approaches. Why isn't this reflected in Table 1?"**
>
> A4: As mentioned in Section 5.1, calculating the Hessian inverse does indeed have a time complexity of $\mathcal{O}(d^2)$. To mitigate this, we use the Hessian-vector product technique in conjunction with conjugate gradient descent, which reduces the overall complexity to $\mathcal{O}(d)$, making it comparable to gradient descent in terms of time complexity, up to constant factors. In Table 1, for clarity, we primarily focus on the cost in terms of the number of iterations, which tends to be large and increases as training progresses. We appreciate your valuable suggestion and will provide further clarification in the next version.
>
> To further assess the impact of estimating the Hessian inverse, we conducted an empirical study comparing the full Hessian inverse with the Hessian inverse computed only for updating the head of the reward model. The results are summarized below:
>
> | Method                   | Accuracy (%) | Training Time (s) |
> | ------------------------ | :-------------: | :-----------------: |
> | Hessian Inverse          | 70.85 ± 0.2   | 3342.9            |
> | HVP Approximation (Ours) | 70.71 ± 0.3   | 1242.3            |
>
> The results demonstrate that our proposed conjunction gradient descent approach can achieve comparable performance to the full Hessian inverse, while significantly reducing the computational cost.
>
> **Q5: "How many seeds did you use for experiments?"**
>
> A5: We used five seeds for each experiment, and we will include this information in the next version.
>
> **Q6: "How did you evaluate the performance of MLE in active collection?"**
>
> A6: To evaluate the performance of MLE in active collection, we use gradient descent on the entire historical dataset during each iteration, aligning with the theory of MLE algorithms in the literature. In contrast, our method only utilizes the current data for optimization in each iteration, significantly improving computational efficiency. This difference highlights the advantage of our approach in terms of reducing the computational burden, as it avoids the need to process the entire historical dataset in every step.
>
> ---
>
> We hope our response addresses your concerns. We are delighted to discuss any further questions you may have.

---

> > ### Comment · Reviewer_mWk5 · 2025-08-01
> >
> > Thank you for your clarification. Happy to raise my rating to 5.

---

> > > ### Author Response · Authors · 2025-08-02
> > >
> > > We thank the reviewer for the positive feedback. We are pleased that our response has helped address the reviewer's concerns, and we will include further clarifications in the revised version accordingly. We sincerely appreciate your valuable feedback!

---

### Official Review · Reviewer_BSpw · 2025-07-02

**Clarity:** 3
**Significance:** 2
**Originality:** 2
**Rating:** 4
**Confidence:** 2

**Summary:**

This paper is about RLHF with an eye towards computational efficiency. Standard methods require solving an optimization problem that has no closed form at each data collection iteration. This paper proposed an OMD-based method where the update in each iteration can be written in closed form. It gives some theoretical guarantees for this method in the linear-reward setting (unregularized regret, no reference policy), with specific guarantees under a few different data collection models -- passive, active, and with deployment time adaptation. It then demonstrates empirical benefits of the proposed method on several benchmarks.

**Questions:**

- In Figure 4a, what method is "Ours (MLE)" referring to?
- It seems that the proposed (theoretical) improvement over the implicit OMD is a factor of log(t) in computational efficiency per iteration. However, the basic convergence guarantee for the proposed method (Lemma 1) pays a factor of log^2(t) in the number of iterations. Is this actually the same # of iterations as the prior methods? Otherwise it's not theoretically apparent whether the overall computational complexity is actually improved.
- Could you explain why the theoretical guarantee in Lemma 1 requires using the second-order approximation rather than first-order, and what the convergence rate would be otherwise?
- The active data collection method claims to avoid the concentrability coefficient between the optimal policy and collected data. However, it still pays a factor of kappa ~ exp(LB). In the related setting where one cares about KL-regularized regret (so the optimal policy is softmax of the reward w.r.t. some reference policy), this kappa seems essentially equivalent to the worst-case concentrability coefficient between all pairs of softmax linear policies with bounded coefficient norms. Thus, active data collection is not required if we're happy with regret bounds scaling with kappa (i.e. it suffices to draw data from the reference policy). Is something similar not true in this paper's setting?
- On a related note, what is the motivation for not regularizing to a base model / reference policy? My impression was that it is desirable to use the base model as a warm start?

**Ethical Concerns:**

["NO or VERY MINOR ethics concerns only"]

**Final Justification:**

The authors' response addressed my main concern about the theoretical results. While they do not entirely match up with the empirical results (due to implementation differences needed to avoid e.g. enumerating over actions, and due to the kappa-dependence which seemingly could be quite bad in the worst case), I think the high-level idea of a principled approach that uses $O(1)$ computation per iteration (and preserves the same theoretical iteration complexity) is an interesting contribution, and the empirical results are promising. Therefore I have increased my score to 4.

**Limitations:**

Yes

**Quality:**

2

**Strengths And Weaknesses:**

**Clarity** The paper is quite well-written, though I have a few clarifying questions (see below). I appreciate the section motivating the design of the proposed OMD method.

**Originality** Prior works such as (Faury et al., 2022) have also tried to improve the computational efficiency of RLHF by simplifying the optimization problem in each individual iteration. The suggested modification that makes each iteration closed-form is to my knowledge novel, though I am not an expert.

**Significance/Quality** The studied problem (computational efficiency of RLHF) is important. The empirical benefits seem fairly convincing, though the theoretical benefits are unclear (see questions).

There are several related works where I would appreciate some discussion on how they relate to the submission:
- The authors mention that recent works [Xie et al., 2025; Cen et al., 2025; Zhang et al., 2025] consider sample efficiency and not computational efficiency. However, there is also the following recent work which aims for both statistical and computational efficiency. The setting is slightly different (KL-regularized regret, and direct reward instead of preference rewards) but I think it is quite related in its goals:

[Foster et al., 2025] "Is A Good Foundation Necessary for Efficient Reinforcement Learning?"

- The following paper achieves a logarithmic regret bound in an RLHF setting; this is equivalent to decreasing the number of iterations necessary (and therefore improving the overall computational complexity, as long as each step is efficient):

[Zhao et al., 2025] "Logarithmic Regret for Online KL-Regularized Reinforcement Learning"

---

> ### Author Rebuttal · Authors · 2025-07-30
>
> Thank you for your helpful comments. We address your main concerns below.
>
> ---
>
> **Q1: "Relationship with Foster et al. (2025)"**
>
> A1: We thank the reviewer for pointing out this related work and will cite it in the next version. While Foster et al. (2025) indeed aim for both statistical and computational efficiency, their approach differs from ours in several key aspects:
>
> 1. **Feedback Mechanism**: Foster et al. (2025) use direct reward signals as feedback, whereas our work focuses on **binary** preference feedback. Consequently, they estimate parameters using a least-squares method, while we develop an OMD-based estimator specifically tailored to preference data, serving as an alternative to the standard MLE estimator.
> 2. **Computational Focus**: Foster et al. (2025) address the computational challenge of *enumerating an exponentially large response space* by leveraging a sampling oracle. In contrast, our work focuses on *mitigating the computational burden that arises from scaling linearly with the number of iterations* by OMD in online RLHF.
>
> In essence, while both works aim to improve efficiency in learning, our contributions are complementary and tackle distinct challenges.
>
> **Q2: "Relationship with Zhao et al. (2025)"**
>
> A2: We thank the reviewer for pointing out this related work and will cite it in the next version. Our work differs from theirs in the following aspects:
>
> 1. They study KL‑regularized contextual bandit problems and achieve a logarithmic regret bound, whereas we focus on the non‑KL‑regularized setting, which typically yields an $O (\sqrt{T})$ regret rate;
> 2. Their work does not address per‑step computational efficiency, for example, the cost associated with Line 5 in Algorithm 1, which is a key component of our contribution.
>
> **Q3: "In Figure 4a, what method is "Ours (MLE)" referring to?"**
>
> A3: We apologize for the confusion. "Ours (MLE)" denotes the variant of our method that employs the MLE estimator for parameter estimation, while still using our proposed action selection strategy as described in Eqs. (6) and (7) to demonstrate its effectiveness. Further details are provided in Lines 298–301, and we will revise Figure 4a to clarify this labeling.
>
> **Q4: "It seems that the proposed (theoretical) improvement over the implicit OMD is a factor of $\log t$ in computational efficiency per iteration..."**
>
> A4: We sincerely thank the reviewer for raising this thoughtful question about the balance between computational efficiency and theoretical guarantees, which indeed gets to the core trade-off of our approach. Importantly, the improvement is not merely per iteration, but also in terms of total computational complexity.
>
> Implicit OMD produces slightly more accurate per-step updates, but at a substantial computational cost: each iteration requires solving an optimization subproblem to sufficient precision, which entails at least **$\mathcal{O}(\log T)$** work per step to avoid a linear regret bound. Our method eliminates this iterative solve entirely by adopting an explicit one-pass update rule, reducing the **per-iteration cost to $\mathcal{O}(1)$**. In effect, we exchange a negligible loss in per-step optimality for a dramatic reduction in computational overhead, which yields a **strict overall runtime improvement**.
>
> As for the $\log^2 t$ factor in Lemma 1: this term appears in the regret bound to control the variance term. In fact, implicit OMD's regret bounds also involve hidden logarithmic factors, while we choose to make ours explicit for clarity. A refined analysis (to be included in the revision) shows that this dependence can be reduced to $\log t$, which is essentially the same as implicit OMD.
>
> In short, the gain is not only at the per-iteration level but also at the level of overall complexity. We will revise the manuscript to make this distinction clearer so that the improvement is fully evident.
>
>
> **Q5: "Why does the theoretical guarantee in Lemma 1 require using the second-order approximation rather than the first-order?"**
>
> A5: Intuitively, the second-order approximation is more accurate than the first-order approximation because it captures the local curvature of the objective function. Technically, the second-order approximation is strongly convex, which introduces a crucial negative term (the last term) in Line 525 of Lemma 2 in the appendix. This term is essential for canceling out non‑essential terms. To the best of our knowledge, using a first-order approximation would invalidate the guarantee and result in a linear regret bound.
>
> **Q6: "concentrability coefficient"**
>
> A6: We are unsure which specific work the reviewer is referencing, but we would like to clarify the distinction in our approach. The worst-case concentrability coefficient can indeed be much larger than $\kappa$, and in some cases, it may be unbounded. For example, when the sampling/reference policy is far from the optimal policy, the concentrability coefficient can grow significantly. This is in contrast to $\kappa$, which represents a more controlled measure in the context of our work.
>
> Therefore, active data collection is necessary to ensure that the regret bound scales with $\kappa$, rather than the potentially much larger worst-case concentrability coefficient. We will add a remark in the next version of the paper to clarify this distinction and further explain why active data collection remains crucial for our approach.
>
> **Q7: "What is the motivation for not regularizing to a base model/reference policy?"**
>
> A7: We recognize that regularizing to a reference policy is a common practice in RLHF. Both approaches, regularizing and not regularizing, have their own advantages and drawbacks. While using a base model as a warm start can be beneficial in certain settings, it may also restrict the flexibility of the learned policy, especially if the reference policy is not well-aligned with the true optimal policy. In our approach, we aim to learn the optimal solution based on the original reward, without being constrained by an imperfect base model. This allows for a more accurate exploration of the problem’s inherent complexities, offering a clearer understanding of its true challenges.
>
> ---
>
> We hope our response has adequately addressed your concerns. We sincerely appreciate your willingness to re-evaluate our work.

---

> > ### Comment · Reviewer_BSpw · 2025-08-02
> >
> > Thanks for the detailed responses addressing my concerns. After taking another look at the paper and the other reviews, I have one additional question -- one of the main empirical strengths of the proposed approach is the runtime compared to MLE.   However, I think I could not locate the implementation details about MLE. If you were to equalize the runtime of MLE and OMD, what would happen to the respective reward model accuracies? I can see from 2(c) that decreasing the number of iterations of MLE would decrease accuracy somewhat; what about decreasing the # of SGD steps per iteration (or is this already only 1)?

---

> ### Author Response · Authors · 2025-08-03
>
> We thank the reviewer for raising this thoughtful point regarding the MLE baseline and runtime comparison. We are glad to clarify the implementation details of MLE and to address the reviewer's equal-runtime concern.
>
> **Implementation details of MLE.** In our experiments, MLE is implemented in the standard way for online RLHF: at each iteration, it optimizes over all historical data collected so far, performing **one full-pass SGD step** per iteration. This choice ensures that MLE benefits from its complete data history and represents a fair and commonly used baseline in online RLHF. By contrast, our OMD method performs **a true one-pass** update, using only the current batch at each step.
>
> **Equalizing runtime.** Following the reviewer's suggestion, we conducted an additional experiment to compare evaluation accuracy across methods under the same wall-clock time budget for the passive data collection stage. We varied the number of SGD steps for MLE per iteration. The results are summarized below:
>
> | Wall Time   | 200s      | 400s      | 600s      | 800s      | 1000s     |
> | ----------- | --------- | --------- | --------- | --------- | --------- |
> | MLE(step = 1) | 67.74     | 68.52     | 69.84     | 69.95     | 70.13     |
> | MLE(step = 5) | 67.25     | 68.10     | 69.32     | 69.55     | 70.01     |
> | **Ours**    | **68.21** | **69.18** | **69.95** | **70.09** | **70.52** |
>
> These results indicate that, under a fixed wall-clock budget, increasing the number of SGD steps per iteration for MLE actually reduces overall accuracy. This occurs because allocating more SGD steps per iteration leaves fewer overall iterations and fewer data samples processed within the same runtime budget, slowing the algorithm's overall progress. In contrast, our proposed method consistently outperforms MLE across all wall-time settings, even when MLE's per-iteration SGD workload is increased. This further validates the efficiency of our proposed method.
>
> We will revise the manuscript to include these results in the next version.

---

> > ### Comment · Reviewer_BSpw · 2025-08-03
> >
> > Thanks for the quick response. This experiment shows that increasing the number of stochastic gradient steps (and decreasing the number of iterations correspondingly) worsens performance. I was asking about the other direction: as I understand, the original MLE baseline, in each iteration, divides the entire history into batches and does a gradient step for each batch -- correct me if I am misunderstanding. However, if I wanted to speed this up, what I would try is to randomly subsample a subset of the history, and only do the batch sgd steps on this subset.
> >
> > Do you know if this would increase or decrease the accuracy, again when equalizing for runtime?

---

> > > ### Author Response · Authors · 2025-08-05
> > >
> > > Thank you for your question. We would like to address your point in three parts:
> > >
> > > 1. **Empirical results on subsampling.** We conducted the experiment you proposed, evaluating MLE with different proportions of randomly subsampled historical data per iteration. The results are summarized below:
> > >
> > >     | Wall Time         | 200s      | 400s      | 600s      | 800s      | 1000s     |
> > >     | ----------------- | --------- | --------- | --------- | --------- | --------- |
> > >     | MLE – 100% Subset | 67.74     | 68.52     | 69.84     | 69.95     | 70.13     |
> > >     | MLE – 50% Subset  | 67.71     | 68.53     | 69.70     | 69.89     | 70.05     |
> > >     | MLE – 20% Subset  | 67.75     | 68.42     | 68.55     | 69.12     | 69.37     |
> > >     | **Ours**          | **68.21** | **69.18** | **69.95** | **70.09** | **70.52** |
> > >
> > >     These results show that **smaller subsets consistently lead to lower accuracy** under the same wall-clock budget. This performance degradation arises because fewer data points are seen and optimized.
> > > 2. **Fundamental differences between MLE and OMD.** While heuristics such as subsampling or reducing the number of SGD steps can help lower MLE's runtime, they **do not resolve the core computational bottleneck** in online RLHF: MLE fundamentally re-optimizes over all prior data at each iteration. In contrast, our OMD method is explicitly designed to perform one-pass updates, which yields a structurally lower per-iteration cost. This is not a tuning detail, but a fundamental algorithmic difference.
> > > 3. **On the value of principled algorithms.** Even if certain heuristic strategies and modifications enable MLE to improve performance and efficiency, this **does not diminish the value of our approach**. Our method supports one-pass reward modeling grounded in theoretical analysis with **provable guarantees**, whereas heuristic adaptations of MLE lack such foundations and may fail to generalize across tasks or conditions.
> > >
> > > We hope this addresses your question and clarifies the significance of our contribution. We would greatly appreciate your reconsideration of our work in light of these points.

---

> > > > ### Comment · Reviewer_BSpw · 2025-08-05
> > > >
> > > > Thanks for running this additional experiment! I find it quite convincing. Independent of the experimental result I do also recognize the value of principled algorithm design, but I would say that the fact that OMD improves upon these basic attempts to scale down computation strengthens the contribution. For these reasons I have increased my score to 4.

---

### Decision · Program_Chairs · 2025-09-17

**Decision:**

Accept (poster)

**Comment:**

This paper considers the online RLHF setting. Previous theoretical works require re-training a reward model at each iteration, which is impractical and time-consuming. Focusing on the linear reward setting, this paper proposes a new single-pass algorithm based on online mirror descent that achieves constant time complexity per iteration. The authors give theoretical guarantees for this approach under data collection models -- passive, active, and with deployment time adaptation, and demonstrate empirical benefits of the proposed method on several benchmarks.

Reviewers generally found the theoretical approach and guarantees to be technically interesting, and were in favor of acceptance after the discussion period. The main drawback on that remains on the theory side is that the algorithm cannot be applied as-is to language models without losing the theoretical guarantees, as it requires enumerating over the action space. This should be discussed in the final version of the paper.